# Convergent neural signatures of speech prediction error are a biological marker for spoken word recognition

Ediz Sohoglu [1] ✉, Loes Beckers [2,3,4] & Matthew H. Davis [2] ✉

We use MEG and fMRI to determine how predictions are combined with speech input in superior temporal cortex. We compare neural responses to words in which first syllables strongly or weakly predict second syllables (e.g., "bingo", "snigger" versus "tango", "meagre"). We further compare neural responses to the same second syllables when predictions mismatch with input during pseudoword perception (e.g., "snigo" and "meago"). Neural representations of second syllables are suppressed by strong predictions when predictions match sensory input but show the opposite effect when predictions mismatch. Computational simulations show that this interaction is consistent with prediction error but not alternative (sharpened signal) computations. Neural signatures of prediction error are observed 200 ms after second syllable onset and in early auditory regions (bilateral Heschl's gyrus and STG). These findings demonstrate prediction error computations during the identification of familiar spoken words and perception of unfamiliar pseudowords.

The information reaching our senses is incomplete and ambiguous such that perception requires active computations for successful interpretation. Nowhere is this more evident than with speech perception. To paraphrase Heraclitus, no listener ever hears the same speech twice, for it's not the same sounds and they're not the same listener. No two productions of the same word are ever acoustically identical; each will be differently realised due to the ubiquitous variability of natural speech produced by different talkers, and in different contexts. No listener is ever the same as they will have different expectations or prior knowledge for the speech they will hear at different times. Consider for example, hearing a word starting with the syllable "tan". It can be unclear whether this is the start of the word "tangle", "tango" or another word that starts with the same initial sounds. This ambiguity can be present even if speech is clearly heard in a quiet background. Yet when listening during everyday conversation, our subjective impression is that spoken word recognition is immediate, clear and accurate. How do listeners achieve such reliable perception despite transient sensory ambiguity?

In answering this question, the notion of Bayesian inference is commonly invoked. That is, listeners combine sensory evidence with prior knowledge about which speech sounds they will hear[1]. This proposal is appealing because it can explain how listeners process speech both rapidly and accurately[2,3]. However, despite widely held agreement that listeners use prior knowledge during speech processing, it remains unclear how this is implemented neurally. One possibility is that neural representations of speech are enhanced by prior knowledge[4]. This has been termed the sharpened signal account[5–7] because in this proposal, neural responses to expected stimuli come to resemble a sharpened (i.e., less ambiguous or less noisy) version of the sensory input. However, it is also the case that expected speech sounds carry relatively little information. Therefore, an alternative and potentially more efficient strategy would be to compute prediction errors, i.e., the difference between heard and predicted sounds[8–11]. In this prediction error account, neural representations are suppressed by prior knowledge and unexpected (and therefore novel and unexplained) sensory information is prioritised for processing.

[1]School of Psychology, University of Sussex, Brighton, UK. [2]MRC Cognition and Brain Sciences Unit, University of Cambridge, Cambridge, UK. [3]Department of Otorhinolaryngology, Donders Institute for Brain, Cognition and Behaviour, Radboud University Medical Center, Nijmegen, The Netherlands. [4]Cochlear Ltd., Mechelen, Belgium. ✉e-mail: E.Sohoglu@sussex.ac.uk; Matt.Davis@mrc-cbu.cam.ac.uk

Sharpened signal and prediction error accounts have been difficult to distinguish experimentally. This is because in both accounts, prior knowledge has a similar effect on the overall strength of neural responses[5,6]. In the case of the prediction error account, brain responses tuned to the heard stimulus features are suppressed when those stimulus features are expected, resulting in an overall reduction in brain activity. With the sharpened signal account, although brain responses tuned to the heard stimulus features are enhanced, this is accompanied by the suppression of responses tuned away from the stimulus input features, i.e., competing interpretations. This suppression of brain responses tuned away from the input features also leads to overall weaker activity. Therefore, previous findings that neural responses correlate with moment-by-moment phoneme and word predictability during story comprehension[12–16] do not provide unambiguous support for prediction error over sharpened signal accounts. Some accounts have proposed that sharpened signal and prediction error computations operate in parallel[8], and localise to different cortical layers[17]. Nonetheless, we focus on contrasting sharpened signal and prediction error mechanisms since single mechanism accounts have been proposed in speech perception[4,18,19] and remain to be distinguished[20].

Multivariate pattern analysis methods that permit measurement of the information conveyed by neural responses can distinguish between sharpened signal and prediction error computations. Indeed, previous work using fMRI[18] and MEG[21] has demonstrated a distinctive neural marker of prediction errors which is that neural representations of sensory features of speech show an interaction between signal quality and prior knowledge. This interaction arises because sensory signals that match strong prior expectations are explained away more effectively as signal quality increases and hence neural representations are suppressed even as perceptual outcomes improve. Whereas for sensory signals that follow less informative prior knowledge, increased signal quality leads to a corresponding increase in sensory information that remains unexplained. By contrast, under a sharpened signal account, computational simulations show that neural representations (like perceptual outcomes) are similarly enhanced by increased signal quality and prior knowledge[18,21]. Therefore, by experimentally manipulating both signal quality and prior knowledge, and observing the consequences on pattern-based neural measures, it is possible to distinguish between accounts.

However, in the aforementioned studies[18,21], listeners heard highly distorted (and unnatural sounding) spoken words while prior knowledge was manipulated by presenting matching or nonmatching written words before speech was heard. This is far removed from real-world conditions in which speech is clearly heard (non-distorted) and predictions are not derived from external written cues but from linguistic knowledge, e.g., word probabilities vary based on usage frequency and preceding sentence context. Indeed, some investigators have argued that previously documented effects of prior knowledge or prediction on language processing are the consequence of prediction encouraging paradigms and fail to generalise to settings more representative of real-world perception[22].

In the current study, we return to this issue by presenting listeners with clearly spoken words. We experimentally manipulated the strength and accuracy of listeners' prior knowledge with the goal of adjudicating between sharpened signal and prediction error accounts. Here our manipulation of prior knowledge is based on information intrinsic to the speech signal and arises from listeners' long-term linguistic knowledge of the sounds of familiar spoken words. We compare neural responses to bisyllabic spoken words (e.g., "bingo", "tango") in which the first syllable strongly (in "bingo") or weakly (in "tango") predicts the form of the second syllable (Fig. 1A). In addition, we compare neural responses to the same second syllables when heard in a pseudoword context (e.g., "snigo", "meago"). Pseudowords, by definition, are unfamiliar to participants and therefore the second

syllable of these items mismatches with listeners' predictions. This creates a listening situation in which strong predictions entirely mismatch with sensory input. This cannot be observed in other experimental situations that forego manipulation of speech content (e.g., story comprehension) and creates a further opportunity for adjudicating between sharpened signal and prediction error computations.

We first performed computational simulations (using model architectures and representations schematised in Figs. 1B, C and S6) to show that, as intended, our experimental manipulations of prediction strength and match/mismatch result in dissociable effects for sharpened signal and prediction error computations (Fig. 1E). While prediction strength and prediction congruency act to enhance sharpened signal representations, prediction error computations show an interaction between prediction strength and congruency. This interaction is driven by a reduction of prediction errors when strong predictions match the sensory input compared with weak predictions. When predictions mismatch with sensory input, however, the opposite occurs with prediction errors slightly increasing with prediction strength. We then measured neural responses using 204-channel MEG and 3T fMRI in separate groups of listeners while they performed an incidental (pause detection) listening task to maintain attention (Fig. S1A). Across multiple imaging modalities (MEG, fMRI), analysis approaches (univariate signal magnitude and multivariate pattern), and signal domains (time and time-frequency), we provide convergent neural evidence that speech perception is supported by the computation of prediction errors in auditory brain regions.

## Results
### Stimulus properties and behavioural responses
To manipulate listeners' predictions, we chose a set of 64 bisyllabic words in which the first syllable (Syl1) either strongly or weakly predicts the same second syllable (Syl2; see Fig. 1A which depicts some example items). Here we operationalise prediction strength as the probability of hearing Syl2, conditioned on Syl1 (p(Syl2|Syl1); see "Methods" section for details and refs. 2,11,12,23.). For example, after hearing the first syllable /b I N/, there is only one potential word candidate (bingo). Therefore, based on their experience of spoken English, listeners will predict that the second syllable /g @U/ is most likely after hearing /b I N/ (p(Syl2|Syl1) = 1). In contrast, after hearing the first syllable "tan", there are more potential word candidates (e.g., tango, tangle, tanker, tankard, tang) and of these words, tango is less frequently used than the competitors tangle and tang. As a result, listeners can only make a weak prediction for /g @U/ after hearing /t {N/ (p(Syl2|Syl1) = 0.052). Note that p(Syl2|Syl1) can also be expressed as syllable surprisal [see ref. 23], which is equivalent to the negative log of p(Syl2|Syl1).

We validated our experimental manipulation of prediction strength by asking a separate group of listeners to perform a free report gating task [see ref. 24 and SI]. This confirmed that listeners were more likely to predict the second syllables of our Strong items than for Weak items. Further validation of our experimental manipulation is provided when assessing listeners' memory for the pseudowords in a cued recall task (see SI).

In addition to manipulating the strength of listeners' predictions, we manipulated whether Syl2 matched or mismatched with listeners' predictions. This was achieved by cross-splicing Syl1 and Syl2 between words to create a set of corresponding pseudowords sharing the same set of syllables as the word items. For real word items, Syl2 always matches predictions, at least to some degree. For pseudowords, p(Syl2|Syl1) approaches zero. This is by definition, since these Syl1 and Syl2 combinations will never previously have been heard by participants as English spoken words prior to the experiment. Therefore, for these items, Syl2 entirely mismatches with listeners' predictions.

It is important to note that even though p(Syl2|Syl1) approaches zero for all Mismatch items, we can still expect a difference in

prediction strength between Strong + Mismatch and Weak + Mismatch items since Strong and Weak items differ not only in terms of p(Syl2|Syl1), but also in entropy over all possible second syllables, i.e., the overall uncertainty of predictions (see SI and Fig. S4A). Indeed in natural speech, conditional probabilities for upcoming speech sounds are negatively correlated with entropy[12,14,23]. Going back to the example above, after hearing the first syllable /b I N/, entropy is relatively low because only one word (bingo) can be predicted. In this case, the unexpected second syllable (/g @ r*/) follows a state of precise

predictions (low uncertainty/entropy). For the word tango on the other hand, entropy is relatively high after hearing the first syllable because predictions are made for multiple second syllables each arising from different word candidates. Here the unexpected second syllable follows a state of imprecise predictions (high uncertainty/entropy). Thus, while we constructed our stimuli based on p(Syl2|Syl1), our manipulation of prediction strength reflects differences in both p(Syl2|Syl1) and syllable entropy. Both aspects of predictability may contribute to perceptual and neural responses in the Match (real word)

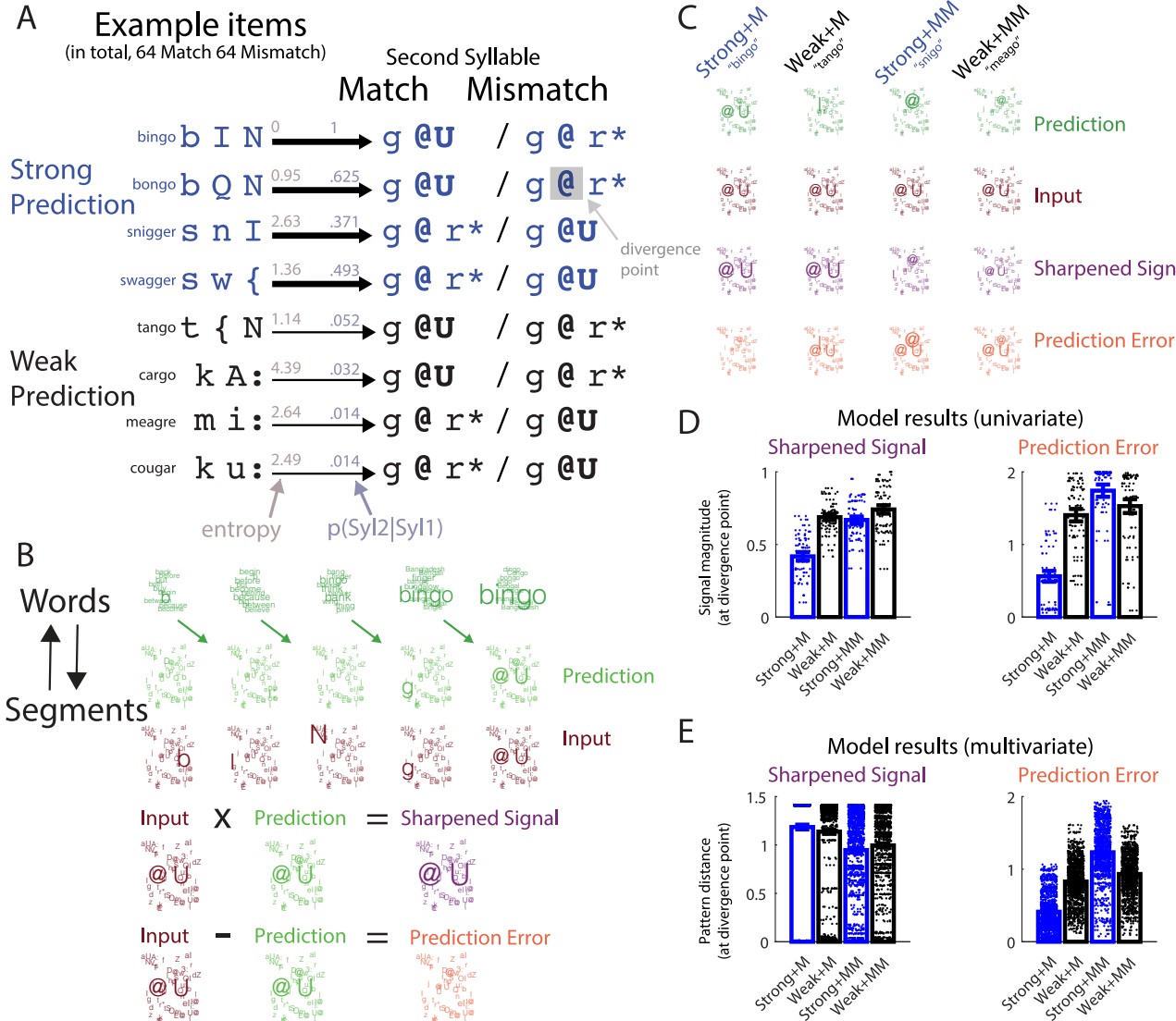

**Fig. 1 | Stimuli and computational modelling. A** Stimuli were bi-syllabic spoken words ("bingo", "bongo", etc. transcribed phonetically as /b I N−g @U/, /b Q N−g @U/) and pseudowords ("bingger", "bongger", etc. transcribed /b I N−g @ r*/, /b Q N−g @ r*/). Strength of prediction for the second syllable (purple) is operationalized as p(Syl2|Syl1) and uncertainty (brown) of predictions is operationalized as entropy (the negative sum of log probabilities). Thickness of arrows depicts higher p(Syl2|Syl1) and lower entropy for Strong vs. Weak Prediction items. The earliest speech segment when the words diverged from the pseudowords (the divergence point) is indicated in bold and always occurred at the second segment of Syl2. **B** Model representations for the word "bingo", visualised using word and segment clouds with larger text indicating higher probabilities (for a more complete illustration, see Fig. S6). (Top row) Word level representations show posterior probabilities as each segment in the input is heard (only the ten words with the highest probabilities at any one timepoint are shown for illustration purposes). Word posteriors provide predictions for which segment will be heard next (second

row, segment predictions shown in green). Segment predictions are then combined with the sensory evidence for individual segments ('Input', third row; shown in brown) through sharpened signal (multiplication) or prediction error (subtraction) computations with outcomes depicted as purple and orange segment clouds, respectively (for the divergence point of "bingo"). **C** Example model representations for four experimental conditions at the divergence point. M Match, MM Mismatch. **D** Results from simulations at the divergence point, used to predict the magnitude of neural responses (univariate signal magnitude analysis). Dots indicate the model signals for each individual item (32 items in each condition) with bars and error bars indicating the mean and standard error across all items.
**E** Results from simulations used to predict neural pattern distances, shown in the same way as (**D**) except the individual datapoints and bars depict the dissimilarity between model representations for all pairs of items (496 item pairs, i.e., 32 × 31/2) within each condition (multivariate pattern analysis).

condition. In the Mismatch condition however, effects of prediction strength can only reflect differences in entropy (as explained above).

To assess neural responses to these stimuli, we recorded MEG ($N = 19$) and fMRI ($N = 21$) data in two separate groups of listeners (Fig. S1A). To maintain listeners' attention, participants were asked to detect brief (200 ms) pauses inserted between Syl1 and Syl2 in occasional target items. As shown in Fig. S1B, C, participants could perform this task quickly and accurately (d-prime higher than 3 and with response times on target-present trials less than 1100 ms).

### Computational simulations

We performed simulations of sharpened signal and prediction error computations supporting spoken word and pseudoword recognition (model architectures and representations schematised in Fig. 1B, C with further illustration of model computations provided in Fig. S6). The underlying mechanisms were derived from established Bayesian models of spoken word recognition (e.g., ref. 2) guided by observations of functionally equivalent but neurally distinct implementations of these. We focus on the timepoint at which the words and pseudowords in our stimuli diverge (the divergence point; highlighted as bold speech segments in Fig. 1A). This is the time at which our experimental manipulation of prediction congruency can affect brain responses: in phonemes, this is at the onset of the second speech segment during Syl2 (98 ms after Syl2 onset on average, with a standard deviation of 46 ms).

Both sharpened signal and prediction error computations are derived from conditional (posterior) word probabilities, as estimated using Bayes theorem (for details, see "Methods"). As with previous work (e.g., refs. 2,11,12), predictions for the next speech segment are estimated from the relative frequencies of the words matching the preceding speech signal, which is appropriate for spoken words heard in isolation (see "Methods" for details). Our estimates of the sensory evidence are derived from the acoustic similarity between individual speech segments and expresses the degree to which the sensory input matches internal representations of spoken words[2,25]. Because our stimuli consisted of clear speech recordings, we simulated conditions in which the sensory evidence was high (i.e., there was minimal sensory uncertainty; see "Methods") but dissociable sharpened signal and prediction error computations are observed when simulating a range of sensory uncertainty levels (Fig. S3).

To relate these simulations to observed neural responses, we apply signal magnitude (univariate) and pattern (multivariate) analysis procedures used in our MEG and fMRI analyses (described in a subsequent section) to simulated representations of speech segments at the divergence point (results shown in Fig. 1D, E). This is performed for each of the four experimental conditions (the 2-by-2 crossing of prediction strength and congruency).

The signal magnitude analysis (Fig. 1D) focusses on the overall strength (rather than pattern) of neural responses. For this magnitude analysis of prediction error representations, we summed model signals (absolute prediction error) over all possible segment representations (48 in total). However, this approach is ill-suited for the sharpened signal simulation since here the model representations correspond to posterior probability distributions and hence all simulated signals are normalised to sum to one (and will hence never differ between conditions). Therefore, to relate sharpened signals to univariate neural responses, we summarised model responses as the normalised sum of log-transformed probabilities over the 48 segment representations (see "Methods"). This choice of linking function between the model and neural responses was motivated by the proposal that neural responses encode log rather than linear probabilities[26]. We also considered an alternative linking function based on entropy, which was motivated by the observation that increased uncertainty in neural responses can be accompanied by an increase in the mean and therefore overall strength of neural responses[9]. The two linking

functions produced equivalent results (this is unsurprising given that entropy is defined as a weighted sum of log probabilities). In the following section we report only the results of using the first linking function (the normalised sum of log-transformed sharpened signals).

This univariate signal magnitude analysis shows a broadly similar pattern for sharpened signal and prediction error computations. For familiar words (Match items), both sharpened signal and prediction error simulations show a reduced univariate response to strongly versus weakly predicted second syllables (Fig. 1D). In addition, univariate responses are overall larger for pseudowords (Mismatch items) versus words (Match items). Within the Mismatch condition, sharpened signal and prediction error simulations diverge somewhat with the former showing a reduced response to strongly versus weakly predicted second syllables and the latter showing a larger response. For both models, prediction strength and congruency have interactive effects on univariate responses as the effect of prediction strength is largest for familiar words (Match items) and reduced for pseudowords (Mismatch items).

The multivariate pattern analysis (Fig. 1E), on the other hand, shows clearer differences between sharpened signal and prediction error computations. This multivariate analysis quantifies how strongly phonetic content in the speech input is represented in simulated patterns at the segment level. This is achieved by quantifying the dissimilarity between model representations (pattern distances) for all pairs of items within each condition.

The results of this multivariate pattern analysis show that while sharpened signal representations are enhanced for Match items, prediction strength and congruency show a cross-over interaction which only occurs for prediction error representations. Specifically, for signals that match prior predictions (during word processing), prediction errors decrease in magnitude and pattern distance for stronger predictions. However, the opposite effect occurs when predictions mismatch with predictions during pseudoword processing; that is prediction error is increased, and more informative for syllables that mismatch with stronger predictions. In addition, prediction errors are overall larger for Mismatch compared with Match items, reflecting the absence of accurate predictions following the divergence point of the pseudoword (Mismatch) stimuli.

These simulations, particularly when analysed using multivariate pattern analysis, confirm that our experimental manipulations can distinguish between sharpened signal and prediction error computations. We can therefore evaluate the extent to which observed brain responses are more consistent with sharpened signal or prediction error computations.

### MEG signal magnitude

As shown in Fig. 2A, we assessed MEG responses timelocked to the onset of Syl2, with respect to a baseline period prior to Syl1 (i.e., speech onset; see "Methods"). Note that as a result, the MEG responses at negative latencies shown in Fig. 2A are non-zero (coinciding with sensory input from the Syl1 portion of our spoken stimuli).

Before assessing the neural timecourse of between-condition differences, we selected 20 (planar gradiometer) sensors with the strongest responses (signal power averaged over conditions, separately for each hemisphere and participant). Sensor selections (i.e., those sensors with the strongest responses) were distributed over temporal and frontal sites bilaterally (shown inset within Fig. 2A), consistent with neural generators in superior temporal cortex. We then summarised MEG responses in these speech responsive sensors in a univariate fashion by taking the RMS amplitude across sensor selections for each timepoint. Reported effects are all FWE (family-wise error) rate corrected for multiple comparisons across timepoints using cluster permutation methods and a clusterwise threshold of $p < 0.05$.

Following Syl2 onset, prediction strength (Strong vs. Weak) and congruency (Match vs. Mismatch) had interactive influences on MEG

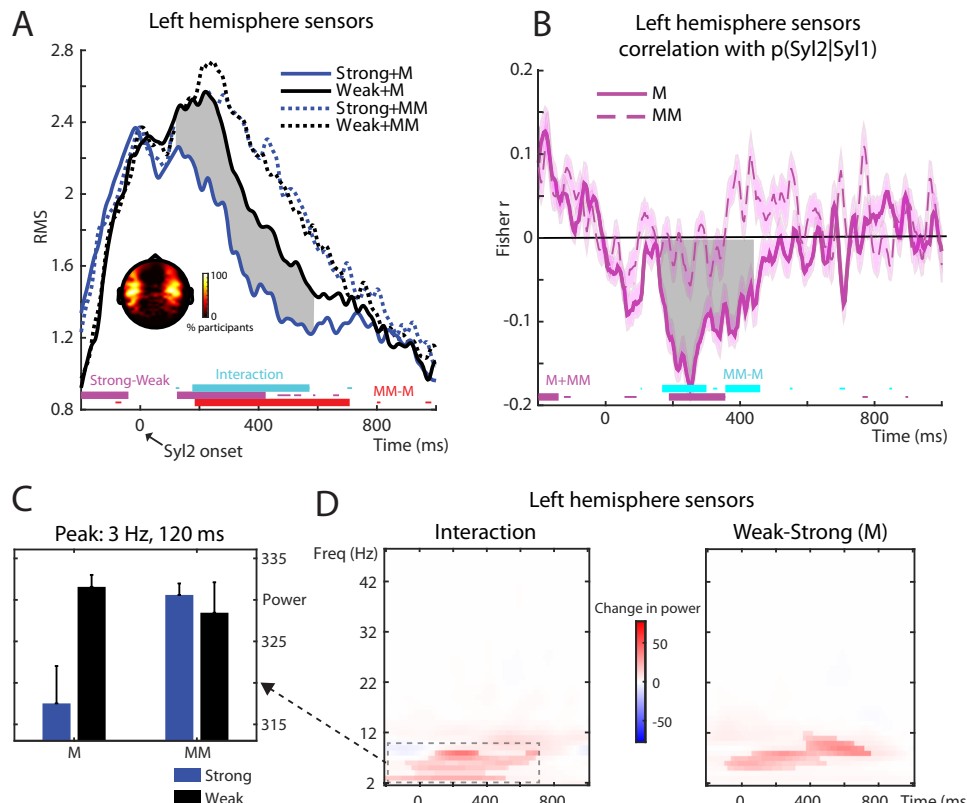

**Fig. 2 | MEG univariate signal magnitude results. A** Traces indicate the root mean square (RMS) of the MEG signal across speech-responsive sensors in the left hemisphere, timelocked to the onset of the second syllable (Syl2). Sensor selections are indicated in the topography as the percentage of participants for which a sensor was selected. Thick horizontal bars at the bottom of the graph indicate timepoints when statistical contrasts (cluster permutation tests) were significant at a cluster-wise threshold of $p < 0.05$ FWE corrected (thin horizontal bars show significance at an uncorrected $p < 0.05$ threshold). Grey shaded area indicates timepoints when the effect of Strong vs. Weak in the Match condition was significant at $p < 0.05$ (FWE cluster corrected). All tests are two-sided. M Match, MM Mismatch. **B** Group-averaged correlations in left hemisphere sensors between p(Syl2|Syl1) and the RMS of the MEG signal. Thick horizontal bars at the bottom of the graph indicate timepoints when statistical contrasts (cluster permutation tests) were significant at a clusterwise threshold of $p < 0.05$ FWE corrected (thin horizontal bars show significance at an uncorrected $p < 0.05$ threshold). Grey shaded area indicates timepoints when the correlations within Match or Mismatch conditions were significant at $p < 0.05$ (FWE cluster corrected). Error bars (transparent area around traces) indicate within-subject standard errors[86]. All tests are two-sided. **C** Bar graph indicates group means ($N = 19$ participants) and within-subject standard errors[86] for the four experimental conditions after averaging MEG power over the temporal and frequency extent of the interaction cluster reported in (**D**). **D** Image depicts changes in MEG power (between condition differences) across time and frequency in left hemisphere sensors for the interaction (left panel) and simple effect of Weak −Strong in Match trials (right panel). Nontransparent colours indicate cluster permutation test significance at $p < 0.05$ (FWE cluster corrected) while transparent colours show $p < 0.05$ (uncorrected). Interaction effect was computed as (Strong-Weak [Mismatch]) − (Strong-Weak[Match]). All tests are two-sided. Source data are provided as a Source Data file.

responses from around 200 to 600 ms (results for left hemisphere sensors are shown in Fig. 2A; similar results are seen for right hemisphere sensors, depicted in Fig. S2A). Stronger predictions resulted in weaker MEG responses but only when speech matched with predictions (significant timepoints for the interaction between prediction strength and congruency are indicated as horizontal cyan bars; the simple effect of Strong vs. Weak for Matching items is shown as the grey shaded area between MEG traces). MEG responses also showed a main effect of prediction congruency with overall larger responses for Mismatch vs. Match items (significant timepoints shown as red horizontal line). These two patterns—interaction between prediction strength and congruency, together with a main effect of Mismatch > Match—are broadly consistent with both sharpened signal and prediction error computations (compare with Fig. 1D). Note that although MEG responses do not increase with prediction strength for mismatching items in this particular analysis (a pattern present only in the prediction error simulations; see Fig. 1D), they do so for other analyses that will be reported below.

Prior to Syl2 onset, a different pattern of results was observed. Here MEG responses increased with prediction strength (indicated as horizontal purple bars in Fig. 2A). While the timing of this effect

(before Syl2 onset) suggests a neural signal reflecting predictions rather than prediction errors, we cannot rule out that this pre-Syl2 effect reflects acoustic differences since the Syl1 portion of Strong items is longer in duration than for Weak items (see SI and Fig. S4A). This duration difference could potentially lead to larger MEG signal when timelocking to Syl2 onset as neural responses would be shifted later in time for Strong versus Weak items. Indeed, when timelocking to Syl1 onset to control for this difference in timing, the increased MEG response for Strong versus Weak items is no longer apparent (Fig. S2C). Note that these or other acoustic confounds can only affect pre-Syl2 neural responses since our cross-splicing procedure ensures that post-Syl2, there are no acoustic differences between our four conditions (see Figs. 1A and S4A). Since representations of Syl2 are critical for distinguishing sharpened signal from prediction error computations (Fig. 1E), we subsequently focus on post-Syl2 effects.

To test whether Syl2 responses are modulated by prediction strength for single items, we correlated p(Syl2|Syl1) with MEG responses across items separately for Match and Mismatch items. We show group-averaged Spearman correlations between prediction strength and MEG responses in left hemisphere sensors in Fig. 2B (right hemisphere correlations shown in Fig. S2B). Consistent with the

previous analysis averaged over items (Fig. 2A), following Syl2, there was an interaction between prediction strength and congruency from 168 to 300 ms and from 356 to 460 ms: Stronger predictions that matched with speech were associated with weaker MEG responses and this correlation differed from that observed when predictions mismatched (significant timepoints indicated as horizontal cyan bars). Individually, the correlations within the Matching condition were significantly negative (indicated as the grey shaded area). Within the Mismatching condition, correlations were weakly positive and significant in the right hemisphere at a later latency (Fig. S2B, 500–600 ms). This pattern of results is more consistent with prediction error than with sharpened signal computations as a positive effect of prediction strength for Mismatch items is only apparent in prediction error simulations (see Fig. 1D). Thus, this analysis demonstrates that the item-averaged effects depicted previously in Fig. 2A extend to single items. That is, the strength of lexical level predictions modulates neural responses to heard speech with the magnitude and direction of this effect depending on whether subsequent segments match or mismatch with predictions. The significant interaction shown in both factorial and graded analyses is broadly consistent with both sharpened signal and prediction error accounts although the positive effect of prediction strength for Mismatch items (observed in graded analysis) favours the prediction error account.

To probe time-frequency correlates of prediction strength and congruency, we also assessed changes in spectral power in the 2–48 Hz range. This time-frequency analysis also showed a significant interaction between prediction strength and congruency in the theta band, with a peak frequency at 3 Hz and a peak latency 120 ms after Syl2 onset (left hemisphere results shown in Fig. 2D; a similar pattern was observed in right hemisphere sensors). Similar to the time-domain analyses above, stronger predictions resulted in weaker MEG power but only when speech matched with predictions (see summary in Fig. 2C). We also analysed high-frequency gamma activity (52–90 Hz) but did not observe reliable condition-wise differences.

## fMRI signal magnitude

In a separate group of listeners, we recorded BOLD functional MRI responses to the same stimuli, using a sparse fMRI sequence so that speech was presented in the silent periods between scans (Fig. S1A). The procedure (task and number of stimulus repetitions) was otherwise identical to that used in the MEG experiment. This allowed us to localise the neural generators of the MEG effects reported above. Reported effects are all FWE rate corrected using a clusterwise threshold of $p < 0.05$.

As shown in Fig. 3A, which presents the univariate fMRI results, the main effect of Mismatch > Match (that was observed following Syl2 onset with MEG) localised to middle/anterior STG bilaterally (see Table 1 for MNI space coordinates). The main effect of Strong > Weak (observed with MEG prior to Syl2 onset) localised to Heschl's gyrus (HG), extending into STG. Despite observing an interaction between prediction and congruency on MEG responses following Syl2 onset, there was no equivalent effect in this fMRI analysis. We suggest this is because of the reduced temporal resolution of fMRI, which makes it more difficult to resolve individual responses to Syl2 and Syl1.

A potentially more sensitive analysis is to correlate prediction strength, i.e., p(Syl2|Syl1) with the magnitude of the BOLD signal across individual items (equivalent to the MEG analysis shown in Fig. 2B). Employing this approach, fMRI responses in bilateral HG and middle STG again increased with prediction strength (positive correlation) for both Match and Mismatch items (shown in Fig. 3B with MNI space

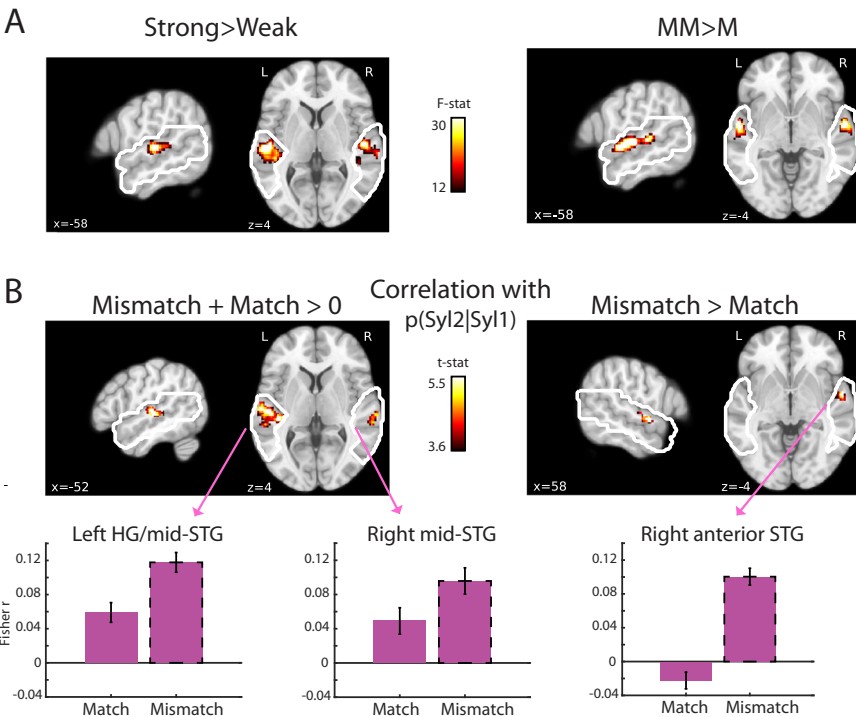

**Fig. 3 | fMRI univariate signal magnitude results. A** Repeated measures ANOVA showed main effects of prediction strength (left panel) and congruency (right panel). Statistical maps ($p < 0.05$ FWE cluster corrected) are shown overlaid onto a template brain in MNI space. White outlines indicate the search volume which included superior and middle temporal regions bilaterally. All tests are two-sided. **B** Statistical maps ($p < 0.05$ FWE cluster corrected) for correlations between p(Syl2| Syl1) and BOLD magnitude. Tests are one-sample *t*-tests (one-sided). In HG extending into middle STG (left panel), correlations are significantly positive and do not differ between Match and Mismatch items (equivalent to a main effect of prediction strength). In anterior STG (right panel), correlations are significantly larger for Mismatch than for Match items. Bar graphs show group-averaged correlations (Fisher-transformed rho-values; *N* = 21 participants) between conditional probability and BOLD magnitude averaged across voxels in the clusters indicated (arrows). Error bars show within-subject standard errors suitable for statistical comparison between conditions. Source data are provided as a Source Data file.

**Table 1 | MNI coordinates for fMRI analyses**

| Location | Extent | Z-stat | x | y | z |
|---|---|---|---|---|---|
| **Univariate Strong > Weak** | | | | | |
| Right Heschl's Gyrus | 551 | 5.74 | 54 | −16 | 2 |
| Right mid-STG | | 4.84 | 64 | −14 | 0 |
| Right mid-STG | | 4.05 | 66 | −26 | 4 |
| Left mid-STG | 551 | 5.41 | −58 | −22 | 4 |
| Left Heschl's Gyrus | | 4.55 | −44 | −24 | 2 |
| Left mid-STG | | 4.55 | −46 | −32 | 4 |
| **Univariate Mismatch > Match** | | | | | |
| Left mid-STG | 569 | 6.00 | −64 | −24 | 6 |
| Left mid-STG | | 5.81 | −60 | −16 | 2 |
| Left anterior-STG | | 5.62 | −58 | −4 | −4 |
| Right anterior-STG | 205 | 5.25 | 58 | −4 | −4 |
| Right mid-STG | | 4.51 | 64 | −16 | 0 |
| Right mid-STG | | 3.54 | 56 | −14 | −2 |
| **Univariate correlations with p(Syl2\|Syl1) Mismatch + Match** | | | | | |
| Left Heschl's Gyrus | 467 | 5.01 | −52 | −22 | 2 |
| Left Heschl's Gyrus | | 4.85 | −42 | −28 | 4 |
| Left mid-STG | | 4.62 | −62 | −18 | 6 |
| Right mid-STG | 105 | 4.04 | 66 | −24 | 6 |
| Right mid-STG | | 3.96 | 64 | −32 | 6 |
| Right mid-STG | | 3.56 | 54 | −26 | 0 |
| **Univariate correlations with p(Syl2\|Syl1) Mismatch > Match** | | | | | |
| Right anterior-STG | 71 | 4.58 | 58 | −2 | −4 |
| Right anterior-STG | | 3.81 | 50 | −4 | −2 |
| Right anterior-STG | | 3.59 | 58 | −10 | 2 |
| **Multivariate Syl2 phonetic distance** | | | | | |
| Right Heschl's Gyrus | 148 | 4.49 | 56 | −6 | 2 |
| Left Heschl's Gyrus | 146 | 4.01 | −44 | −20 | 0 |

Also shown are the spatial extent (number of voxels) and peak statistic for each cluster.

coordinates listed in Table 1), demonstrating that the effect depicted previously for items that differ in averaged prediction strength in Fig. 3A is also observed for single items which encompass a range of prediction strengths. Because this effect of prediction strength is positive and equally apparent for Match and Mismatch items, we suggest it corresponds to the pre-Syl2 effect observed in MEG, i.e., reflecting processing prior to the critical Syl2 event. We therefore will not discuss this effect further because our focus is post-Syl2 processing (as explained above), which can distinguish different computational accounts.

Importantly, we also observed an interaction between p(Syl2|Syl1) and prediction congruency in right anterior STG (shown in Fig. 3B), revealing a potential neural generator of the interaction effects previously observed with MEG after Syl2 onset (Figs. 2B and S2B). Although this fMRI effect appears driven by a positive correlation in the Mismatch condition (rather than a negative correlation for Match items), the correlation is numerically negative for Match items, which is consistent with earlier MEG analysis (Figs. 2B and S2B). We also note that like the present fMRI analysis, right-hemisphere MEG sensors show a positive correlation in the Mismatch condition at later latencies (Fig. S2B). As noted earlier for the univariate MEG analysis, this positive effect of prediction strength for Mismatch items is only apparent in prediction error simulations (Fig. 1D).

### MEG pattern analysis

The univariate signal magnitude analyses above shows that the magnitude of neural responses in superior temporal regions is modulated by syllable prediction strength in different directions depending on whether heard speech matches or mismatches with predictions. While the direction of these neural effects is in line with the magnitude of prediction error in our computational simulations, as mentioned above, our computational simulations indicate that multivariate pattern analysis more clearly distinguishes sharpened signal and prediction error accounts. Simulations of prediction error computations show that representational patterns show a specific cross-over interaction between prediction strength and congruency (Fig. 1E) which is clearly absent for sharpened signal computations.

To examine how predictability modulates neural representational patterns in our MEG data, we computed the Euclidean distance between sensor patterns (over the entire 204 array of planar gradiometers) evoked by pairs of items. Taking this approach, we first asked whether the MEG signal contained phonetic information about the syllables by correlating neural pattern distances with the phonetic dissimilarity between the syllables of item pairs (based on a representational dissimilarity matrix derived from the Levenshtein distance between phonetic transcriptions for different syllables, as illustrated in Fig. 4A for Syl2 and Fig. S5 for Syl1). As shown in Fig. 4B, MEG patterns correlated with Syl2 phonetic dissimilarities only after Syl2 onset (reliable effects were observed in two clusters between 168 and 928 ms post-DP, with the peak effect observed at 332 ms; indicated by horizontal red bars). In contrast the peak correlation with Syl1 phonetic dissimilarities was largely confined to periods in which Syl1 was heard, with a cluster from −200 to +188 ms, and peak effects occurring at −112 ms (indicated by horizontal beige bars). This confirms that MEG response patterns represent the phonetic content of the syllables as listeners are hearing them, in line with other, similar demonstrations[27]. Qualitatively similar albeit weaker results are obtained when using cross-validated Mahalanobis distances[28].

We next assessed how neural pattern distances were modulated by syllable predictability by averaging pattern distances over item pairs within each condition and comparing averaged distances between conditions [for a similar approach applied to visual processing, see ref. 29]. As shown in Fig. 4C, following Syl2 onset, prediction strength and congruency interacted to influence MEG pattern distances: stronger predictions resulted in larger MEG pattern distances but only when speech mismatched with predictions (simple effect of Strong−Weak [Mismatch] is shown as shaded grey area in Fig. 4C, and the first interaction cluster peak at 236 ms, is plotted as an inset). This interaction revealed for MEG patterns after Syl2 onset is more consistent with prediction error than with sharpened signal computations (compare with Fig. 1E).

Complementing the analysis above, we also examined the correlation between neural pattern distances and Syl2 phonetic dissimilarities, separately for each condition (i.e., the same dissimilarities indicated in Fig. 4A but within condition). This provides a second measure of neural representations, one that is potentially more specific to Syl2 phonetic content as opposed to other properties that could differ between items e.g., acoustic, lexical, semantic etc. However, we did not find any timepoints showing an interaction between prediction strength and congruency nor any main effects (Fig. S2D). Given that our simulations provide an apriori hypothesis that the effect of Strong vs. Weak predictions should differ for Match and Mismatch conditions (see Fig. 1E), we also tested for simple effects of Strong versus Weak prediction within each congruency condition. This revealed greater correlation between MEG pattern and Syl2 phonetic distances for Strong versus Weak items in the Mismatch condition only. This occurred in two clusters, from 44 to 116 ms and from 148 to 220 ms. These results remain broadly consistent with the earlier analysis shown in Fig. 4C. While here we did not observe an interaction between prediction strength and congruency, as before we did observe a positive effect of prediction strength within the Mismatch condition. This result favours the prediction error account (see Fig. 1E).

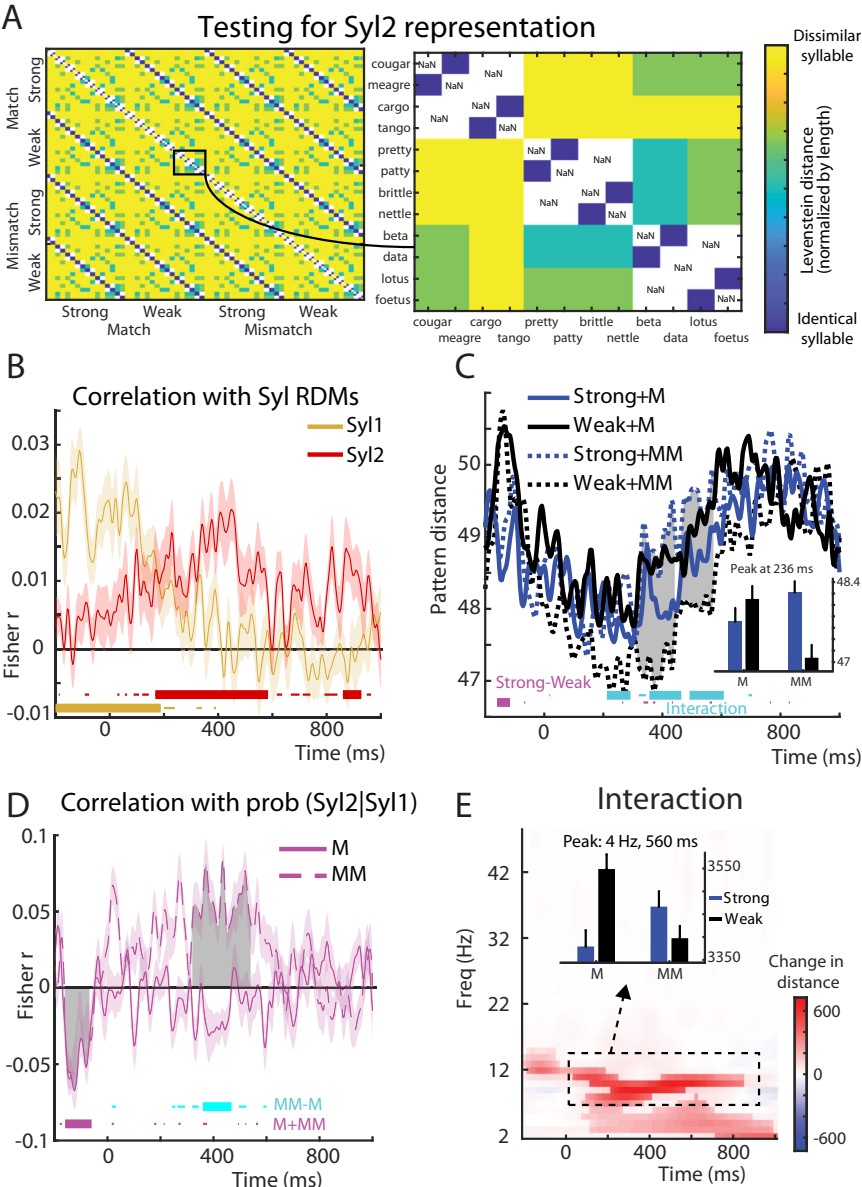

**Fig. 4 | MEG multivariate pattern analysis results. A** Phonetic dissimilarity matrix expressing the normalised Levenshtein distance between phonetic transcriptions of Syl2 for different item pairs. Left panel shows dissimilarities for all items while right panel highlights dissimilarities for a subset of items. **B** Group-averaged correlation between MEG pattern distances and phonetic dissimilarities for Syl2 (in red) and Syl1 (in beige). Horizontal bars indicate timepoints when correlations were significantly greater than zero (thick lines show cluster permutation test significance at $p < 0.05$ FWE cluster corrected; thin lines show uncorrected significance at $p < 0.05$). Error bars (transparent area around traces) indicate standard error of the mean. Tests are one-sided. **C** Group-averaged MEG pattern distances with larger values indicating more dissimilar MEG response patterns between items. Thick horizontal bars at the bottom of the graph indicate timepoints when statistical contrasts (cluster permutation tests) were significant at a clusterwise threshold of $p < 0.05$ FWE corrected. Thin horizontal bars show significance at an uncorrected $p < 0.05$ threshold. Grey shaded area indicates timepoints when there was a significant difference between Strong and Weak items in the Mismatch

condition ($p < 0.05$ FWE clusterwise corrected). The inset bar plot shows group means for each condition ($N = 19$ participants) and within-subject standard errors, after averaging over the temporal extent of the cluster for the interaction contrast. All tests are two-sided. **D** Group-averaged correlations between p(Syl2|Syl1) and MEG pattern distances. Grey shaded area indicates timepoints when there was a significant correlation for Mismatch items ($p < 0.05$ FWE clusterwise corrected). Error bars (transparent area around traces) indicate within-subject standard errors[86]. All tests are two-sided. **E** Group-averaged MEG pattern distances after transforming the MEG signal into time-frequency power. Image depicts the interaction contrast: (Strong-Weak [Mismatch]) – (Strong-Weak[Match]). Non-transparent colours indicate significance at $p < 0.05$ (FWE cluster corrected) while transparent colours indicate $p < 0.05$ (uncorrected). Bar graph indicates group means ($N = 19$ participants) and within-subject standard errors, after averaging over the temporal and frequency extent of the interaction cluster (black broken outline on the time-frequency image). All tests are two-sided. Source data are provided as a Source Data file.

Correlating pattern distances with a graded measure of syllable prediction strength (i.e., p(Syl2 | Sy1)) revealed further, convergent evidence for prediction error computations (Fig. 4D). Stronger predictions for upcoming speech sounds during Syl2 were associated with opposite effects on MEG pattern distances from 360 to 468 ms, depending on prediction congruency (significant timepoints indicated

as horizontal cyan bars). During this time period, correlations between prediction strength and pattern distances were positive when predictions mismatched with heard speech (i.e., neural representations were more informative for speech that mismatched with stronger predictions), and negative when predictions matched heard speech (i.e., less informative representations for speech that matched stronger

predictions). These findings are as expected for representations that signal prediction error (Fig. 1E right). Individually, correlations within the Matching condition were not significantly different from zero. But within the Mismatching condition, correlations were significantly positive from 320 to 544 ms (shown as shaded grey area in Fig. 4D).

We also applied our pattern analysis procedure to MEG sensor-space patterns after the data was transformed into a time-frequency representation of signal power (Fig. 4E). Here there was again an interaction between prediction strength and congruency with a peak at 560 ms in the theta band (peak at 4 Hz, extending to higher alpha frequencies). This interaction was driven by smaller MEG pattern distances (i.e., less informative neural representations) for Matching syllables that follow Strong rather than Weak predictions (Fig. S2E). Again, this effect is in line with neural responses that signal prediction errors in heard speech. We also applied the same pattern analysis approach to high-frequency gamma activity (52–90 Hz) but did not observe condition-wise differences.

## fMRI pattern analysis

To localise the neural generators of the MEG pattern analysis effects reported above, we applied a similar multivariate pattern analysis approach to spatial searchlights in auditory brain areas in our fMRI data. As with our MEG pattern analysis, we first asked whether fMRI responses contained phonetic information about Syl2 by correlating neural pattern distances with the phonetic dissimilarity between the second syllables of item pairs. As shown in Fig. 5A, multivoxel patterns correlated with Syl2 phonetic dissimilarities in HG bilaterally (see Table 1 for MNI space locations).

We next proceeded to examine between-condition differences in neural representations. Rather than directly analysing pattern distances (as with the equivalent MEG analysis shown in Fig. 4C), we opted to analyse a more specific measure of the correlation between neural patterns and Syl2 phonetic dissimilarities. This approach is more optimally suited to measure fMRI responses that are specific to Syl2 as opposed to representing both syllables. This is because while the

temporally precise MEG signal can easily resolve neural responses to Syl1 and Syl2, this is not the case for fMRI which integrates sensory information over both syllables. However, by analysing the strength of correlation between fMRI pattern distances and Syl2 phonetic dissimilarities, we can more specifically examine the neural representations of Syl2. Within the left and right HG clusters previously shown in Fig. 5A, prediction strength and congruency showed interactive effects on the correlation between multivoxel pattern distances and Syl2 phonetic dissimilarities (shown in Fig. 5B; $F(1,20) = 10.0$, $p = 0.005$, $\eta^2_p = 0.333$). This effect did not reliably differ in magnitude between the two hemispheres (the three-way interaction between prediction strength, congruency and hemisphere was non-significant: $F(1,20) = 0.170$, $p = 0.684$, $\eta^2_p = 0.333$). Though the interaction effect failed to reach significance in the left hemisphere ($F(1,20) = 3.30$, $p = 0.084$, $\eta^2_p = 0.142$), the interaction effect was significant in the right hemisphere ($F(1,20) = 7.83$, $p = 0.011$, $\eta^2_p = 0.281$). While tests of prediction strength within each congruency condition and hemisphere were not reliable, the numerical trends over hemisphere are entirely consistent with previous analyses and in line with computational predictions: increasing prediction strength resulted in a numerically weaker correlation for a matching Syl2 ($F(1,20) = 2.85$, $p = 0.107$, $\eta^2_p = 0.125$) while the opposite numerical difference (i.e., a stronger correlation for stronger predictions) was observed for mismatching Syl2 items ($F(1,20) = 2.70$, $p = 0.116$, $\eta^2_p = 0.119$).

We also examined whether fMRI responses contained phonetic information about Syl1 by correlating neural pattern distances with the phonetic dissimilarity between the first syllables of item pairs. However, we did not find significant correlations (all clusters $p > 0.6$ uncorrected).

## Discussion

Across multiple imaging modalities (MEG, fMRI), analysis approaches (univariate signal magnitude and multivariate pattern), and signal domains (phase-locked, evoked activity and induced, time-frequency MEG responses), we show that neural responses to second syllables in spoken words are modulated by the strength and content of prior knowledge (i.e., predictions for expected speech sounds). We further show that these neural influences of prior knowledge begin around 200 ms after the onset of the second syllable, localise to early auditory regions (in fMRI, bilateral Heschl's gyrus and STG) and are also expressed as changes in low-frequency (theta and alpha) power.

### Neural patterns distinguish prediction error from sharpened signal computations

In the current study, the critical analyses used multivariate pattern analysis procedures that previous work has shown can unambiguously distinguish between sharpened signal and prediction error accounts[18,21]. By examining the pattern and not just the overall strength of neural signals, responses tuned to the stimulus input features as well as those tuned away from the heard stimulus can be jointly modelled. In this analysis framework, if neural response patterns reflect prediction errors they should contain less information about strongly versus weakly predicted speech sounds (see computational simulations in Fig. 1E), and this is indeed what we observed for MEG and fMRI responses to familiar words.

While neural representations of second syllables are suppressed for stronger predictions that match heard speech, neural representations of exactly the same syllables show the opposite modulation for second syllables of pseudowords. That is, when predictions mismatch with sensory input, neural representations are enhanced after more strongly- than more weakly-predicting syllables. Our computational simulations and multivariate analysis confirm that the interaction between prediction strength and match/mismatch is entirely consistent with the operation of prediction error computations[8–11] but cannot be explained by sharpened signal accounts in which

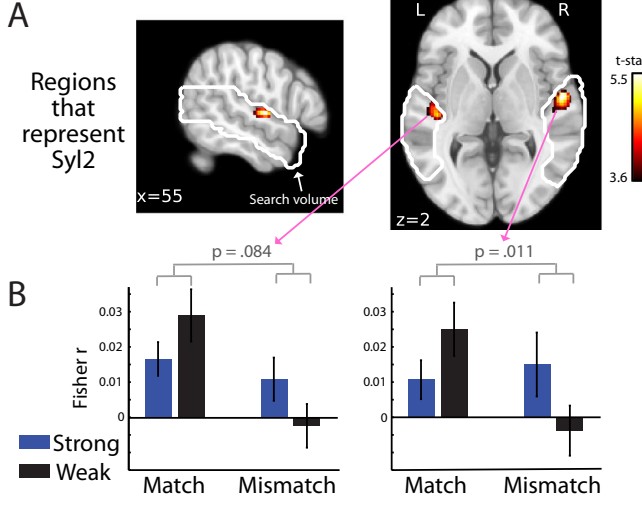

**Fig. 5 | fMRI multivariate pattern analysis results. A** Statistical maps showing voxels in which there was a significant correlation between fMRI pattern distances and phonetic dissimilarities for Syl2 ($p < 0.05$ FWE corrected clusterwise). Tests are one-sample $t$-tests against zero (one-sided). White outline shows search volumes as in Fig. 3. **B** Group-averaged correlations ($N = 21$ participants) and within-subject standard errors, after averaging over voxels in the left and right hemisphere clusters shown in (**A**). $p$ values are for the two-way interactions between prediction strength and congruency from repeated measures ANOVA. All tests are two-sided. Source data are provided as a Source Data file.

representations of speech are enhanced by prior knowledge[5–7]. Our study therefore provides convergent neural evidence—from different imaging modalities and multiple analysis approaches—that speech perception is supported by the computation of prediction error representations in auditory brain regions.

It should be noted that in prediction error accounts, computation of prediction errors is not the end goal of the system (as discussed in ref. 21). Rather, they are used to update—and thereby enhance or sharpen—future predictions. In some formulations of predictive coding[8,17], these sharpened predictions are represented in separate neural populations. However, we saw little evidence of sharpened representations in our study; perhaps because these sharpened predictions reside in deeper cortical layers[17] to which MEG[30] and BOLD[31] signals are less sensitive. It has also been proposed that prediction errors and predictions are communicated via distinct oscillatory channels[32,33], although our time-frequency analysis did not provide support for this (discussed below). In future, methodological advances, e.g., in layer-specific imaging[34], or more invasive methods[35] may help to resolve this issue. Insights may also come from studies that simultaneously probe multiple levels of processing (e.g., to observe prediction updating in higher-level semantic stages of processing).

Why are prediction error representations enhanced by prediction strength when predictions mismatch with speech input i.e., for pseudowords? Because the combinations of syllables in these items will never previously have been heard by listeners as English spoken words, the expected probabilities for the second syllables should always approach zero (in information theory terms, surprisal should be maximal; see ref. 3). Therefore, if neural representations were only sensitive to predictions for the heard syllables (and not alternative syllables), we would not expect any neural differences between Strong + Mismatch and Weak + Mismatch items. However, our manipulation of prediction strength reflects not only differences in predictions for the heard second syllables, but also in the uncertainty of predictions over all possible syllables: In information theory, this uncertainty is formulated as entropy[23]. Indeed in natural speech, predictions for specific speech sounds are negatively correlated with entropy[12,14,23].

We suggest our results reflect both aspects of prediction, i.e., for the heard syllables as well as the uncertainty of predictions over all possible syllables. In the Strong + Mismatch condition, a focal (low entropy) prediction is misaligned with sensory input. As a result, there is increased information in neural prediction errors because the sensory signal remains unexplained. By contrast, weak and more distributed (lower entropy) predictions will tend to align more with speech input representations. This illustrated by the pseudoword tanger. After hearing the first syllable /t {N/, predictions are spread weakly over multiple word candidates including the word *tanker* in which the penultimate sound is shared with tanger. This overlap results in reduced prediction errors in the Weak + Mismatch condition.

Our findings are consistent with neurophysiological findings showing that in addition to the informativeness of heard speech sounds, brain responses are also sensitive to the uncertainty of predictions made in parallel for alternative sounds[12,14,27,36]. Going beyond this previous work however, our current findings suggest that neural sensitivity to prediction uncertainty reflects the specific neural computation of prediction error. Prediction uncertainty is a parsimonious explanation of the prediction strength effect on pseudoword representations. However, it is also possible that the strong violations of predictions that occur for pseudoword items may engage additional or qualitatively different mechanisms to those that support processing of familiar words. Bayesian inference for unfamiliar words is ill-defined because a pseudoword by definition has never previously been heard and therefore has a prior probability of zero[2,3]. In this situation, prediction error is maximal[3] and may function as a signal to indicate that the listener's existing internal model is no longer applicable[3,37], triggering the encoding of a new word that may later enter into the lexicon (see below for further discussion and refs. 3,38) or perceptual learning if the speaker systematically produces certain speech sounds with an unusual pronunciation[39]. By this view, opposite neural effects of prediction strength during perception of matching and mismatching second syllables arise because distinct neural processes are engaged during perception of words and pseudowords.

## Prediction error computations during natural listening

Our findings replicate and extend the findings of previous studies that manipulated prior knowledge by presenting matching or nonmatching written words before highly degraded spoken words[18,21,40]. In this previous work, neural representations showed interactive influences of prior knowledge and signal quality, consistent with a prediction error account. We go beyond this previous work by demonstrating prediction error computations in a more natural listening situation. The present experiments are more naturalistic in two respects: Firstly, we presented clear speech rather than the highly degraded (and hence less natural sounding) noise-vocoded words used previously. Secondly, we used an intrinsic manipulation of prior knowledge based on listeners' long-term linguistic knowledge of familiar spoken words rather than an extrinsic manipulation of predictions based on written text. This demonstration of prediction errors during more natural listening situations is important because it directly addresses the concern that previous results reflect a prediction encouraging paradigm[22] that would not generalise to other listening situations. Instead, our findings are consistent with the idea that lexical identification operates through prediction error computations and that these are integral to perceptual processing of speech irrespective of the listening situation[3,41–44].

One might argue that our findings fall short of demonstrating prediction error computation in a fully naturalistic listening situation as our stimuli comprised individual spoken words as opposed to connected speech. In other work investigating speech processing during perception of connected speech (e.g., stories), neurophysiological responses (measured by MEG, EEG or electrocorticography) are shown to correlate with various measures of time-varying predictability, including phoneme surprisal[12–15,45], phoneme entropy[14,45] and lexical entropy[12,15,45]. While these findings clearly demonstrate that speech processing is strongly modulated by prior knowledge, consistent with the notion that speech perception is fundamentally a process of Bayesian inference[2], they do not indicate whether the underlying neural computations involve sharpened signals or prediction errors. Consider for example, the common observation of reduced responses to predictable speech sounds. This response reduction is compatible with either a prediction error account (through the suppression of responses tuned to input features that are predicted) or a sharpened signal mechanism (through the suppression of responses tuned away from the input features; for a more detailed explanation, see Introduction and refs. 5,6,20). Indeed, this pattern is evident in our simulations as univariate responses are reduced for strongly versus weakly predicted second syllables of words for both sharpened signal and prediction error simulations.

More recently, studies employing connected speech paradigms have shown that neural representations of speech segments and acoustic features are themselves modulated by predictability[16,27,46]. These studies use an approach analogous to the present pattern analysis since they specifically link neural effects of surprisal and entropy to neural representations of specific segments or acoustic features. However, previous findings appear inconsistent with respect to sharpened signal and prediction error accounts since some studies show that neural representations are enhanced when predictions are strong[46] while others show the opposite[16] or even effects in both directions at different latencies[27].

These inconsistent results from story listening paradigms therefore suggest that alternative experimental manipulations of listeners' predictions are needed to distinguish between accounts. Our pseudoword condition creates a second situation in which prediction error and sharpened signal computations can be teased apart. Pseudowords are previously unfamiliar to participants, and hence create a unique listening situation in which predictions entirely mismatch with sensory input. Under a sharpened signal account, the pattern of neural representations of speech sounds forming a pseudoword (as compared with a word) should be suppressed, reflecting the low prior probability of hearing these speech sounds. In contrast, under a prediction error account, the exact opposite should occur i.e., enhanced representations for pseudowords, particularly following strongly-predicting first syllables. Neural responses to pseudowords are consistent with this latter pattern and hence are likely to reflect prediction error computations. In our signal magnitude analyses, we observed larger MEG and fMRI responses for pseudowords versus words. For pattern analyses, although we did not observe a reliable main effect of prediction congruency, neural representations of speech for pseudoword items were enhanced following strongly- versus weakly-predicting first syllables. Overall, these pseudoword results again favour a prediction error account.

We have previously argued that larger prediction error responses for pseudowords compared to real words provide a potential signal for novelty detection and pseudoword learning (ref. 3, interpreting results from refs. 11,47). Our demonstration of more informative prediction errors for syllables that mismatch with more strongly-predicting first syllables suggests that such pseudowords might be more readily learned. We note that memory studies using visual stimuli have similarly argued for enhanced learning of face-scene associations that generate larger prediction errors at encoding[48]. Consistent with this proposal, are the present behavioural results from the cued memory recall test conducted after the fMRI scans. This showed that listeners' recall for pseudowords is more accurate for items that strongly mismatched with predictions in the scanner.

## Spatiotemporal and spectral profile of prediction error computations

Our results also establish the timing and location of prediction error computations in response to speech. Previous studies using EEG[13], MEG[12–14,27], fMRI[18,49–52] and electrocorticography[15,53] disagree about when and where prediction errors are computed in the cortical hierarchy. Each of these techniques has their strengths but none combines spatiotemporal resolution with wide coverage of the cortex. Perhaps because of this, it remains unclear whether predictive neural computations emerge only once sensory information has been transformed into non-acoustic (e.g., phonetic) representations[12] or whether acoustic representations themselves can support predictive computations[21]. In the present study, we obtained MEG and fMRI responses to the same stimuli, while participants performed the same incidental (pause detection) task to maintain attention. This feature of our study enabled us to map out prediction error computations precisely in both time and space.

In our MEG results, effects of prediction strength and congruency occurred around 200 ms following the onset of the critical second syllable. Note however that our word and pseudoword stimuli are identical until approximately 100 ms after the onset of the second syllable (corresponding to the onset of the subsequent segment). Therefore, relative to when the critical acoustic information became available to listeners, the interaction between prediction strength and congruency occurred with a latency of approximately 100 ms. This latency is compatible with a low-level acoustic locus of prediction error computation but perhaps not early enough to definitively rule out a higher-level (non-acoustic) locus, if word identification is supported by rapid (<50 ms) cortical mechanisms[54].

However, our fMRI results enable more definitive conclusions, since searchlight pattern results show prediction effects in primary auditory cortex (Heschl's Gyrus). This localisation to Heschl's Gyrus is striking as it suggests that predictive computations for speech can operate even at the lowest levels of the cortical processing hierarchy, which leaves open the possibility that top-down predictive influences may even extend subcortically[55]. Future work manipulating prediction at different linguistic levels (e.g., sublexical, lexical and supralexical) would be valuable to establish the generality of this finding, and to determine whether these effects are uniquely or differentially localised to specific cortical regions.

In both signal magnitude and pattern analysis of MEG responses, the interaction between prediction strength and congruency occurred in low-frequency neural responses, with a peak in the theta range (3–4 Hz). An influential proposal claims that bottom-up prediction errors are conveyed by high-frequency (gamma) activity while top-down predictions are conveyed by low-frequency (alpha/beta) signals[17,32,56]. This is consistent with the idea that neural representations integrate sensory information over longer timescales at higher cortical levels (where top-down predictions are proposed to originate; see refs. 49,50,57–59). If the current findings reflect prediction errors as we have argued, then we might have expected modulation in the gamma range, which we did not observe. Instead, our findings are consistent with other work suggesting that prediction errors for speech are conveyed in low-frequency neural signals[14,60]. This does not rule out an additional role for gamma responses. It may be for example that theta and gamma responses are coupled[61,62] and that gamma effects are not observed in the present study due to the generally lower signal-to-noise ratio of MEG responses at higher frequencies[63].

## Integrating computational models and brain responses to reveal the neural computations supporting speech processing

The current study follows previous work that also combined computational models and neuroimaging to test sharpened signal and prediction error accounts[18,21]. As has been commented on elsewhere[64], computational models provide the most reliable way to establish the predictions of a particular theory. This observation also applies to the goal of distinguishing sharpened signal and prediction error theories. For example, opposite outcomes are observed in simulated prediction error representations depending on the level of sensory degradation (see refs. 18,21 and also computational simulations in Fig. S3 and "Methods" section). While this result is clearly apparent in simulations, it was not previously obvious from verbal descriptions of prediction error theories. In the current study, we also see opposite outcomes in simulated prediction errors depending on whether predictions match or mismatch sensory input during word and pseudoword processing, respectively. Through computational simulations, we could establish that this is a unique signature of prediction error representations during perception of words and pseudowords and furthermore show that this signature is observed in neural responses.

Importantly, our simulations also provide an explanation for why prediction strength and match/mismatch combine interactively to influence prediction error representations (see discussion above on why prediction error representations are enhanced by prediction strength when predictions mismatch speech input). However, future work is needed to develop a more complete model of prediction error processing (for examples of other ongoing efforts, see refs. 65–67). For example, in the present model implementation, predictions for upcoming speech sounds are subtracted from the input to compute prediction errors but these segmental predictions are never used to update higher-level lexical predictions. Rather, lexical-level predictions are computed afresh as each new segment is presented (by combining word priors and likelihoods using Bayesian inference). A more complete model would update predictions via prediction errors. It would also operate on real (i.e., acoustic) speech input or connected

speech without explicitly requiring word onset locations. We note with interest the success of recent computational models which operate on acoustic speech inputs and are used to predict neural data[68,69]. We hope that the present results can constrain future iterations of these models.

## Methods

### Participants

Forty participants (19 in the MEG experiment, 21 in the fMRI experiment) were tested after they gave informed consent and were informed of the study's procedure, which was approved by the Cambridge Psychology Research Ethics Committee. Participants were recruited from the volunteer research panel of the MRC Cognition and Brain Sciences Unit. Participants received a monetary inconvenience payment for their participation (£10 per hour and travel expenses). All were right-handed, native speakers of English, aged between 18 and 40 years and had no self-reported history of hearing impairment or neurological disease. The mean age in the MEG experiment was 25 years (SD = 4.63; 12 female, 7 male) and 23 years in the fMRI experiment (SD = 3.14; 13 female, 8 male). We aimed to recruit both females and males without special consideration to matching their number. Gender status was based on self-report.

### Stimuli

Spoken stimuli were 16-bit, 44.1 kHz recordings of a male speaker of southern British English and their duration ranged from 388 to 967 ms (mean = 621, SD = 107). A set of 64 bisyllabic words (Match items) were first selected from the CELEX database[70] based on the probability of the second syllable (Syl2) conditioned on the first syllable (Syl1):

$$p(\text{Syl2}, |, \text{Syl1}) = \frac{\sum \text{freq(words matching both syllables)}}{\sum \text{freq(words matching first syllable)}} \quad (1)$$

For this we used syllabified phonetic transcriptions in CELEX such that the first syllable /t {N/, for example, in words such as tango and tangle were considered to match the speech signal since all these words share the same first syllable /t {N/. However, words such as "tank" were not considered as matching since this word consists of the single syllable /t {N k/. Words were selected as sets of eight items (one example item set is shown in Fig. 1A). Within a set of eight items, each word had a unique Syl1 that strongly or weakly predicted one of two Syl2 syllables. By cross-splicing the first and second syllables of these words, we were able to create a set of pseudowords (Mismatch items) that contained the same Syl2 syllables as the real words.

To ensure that Syl2 syllables shared between items were acoustically identical, all four recorded syllables were combined using audio morphing using STRAIGHT software[71] implemented in Matlab (The Mathworks Inc). This software decomposes speech signals into source information and spectral information, and permits high quality speech resynthesis based on modified versions of these representations. This enables flexible averaging and interpolation of parameter values that can generate acoustically intermediate speech tokens[47],for application see ref. 72. Exploiting this capability of STRAIGHT, all words and pseudowords sharing the same Syl2 were constructed to be acoustically identical following Syl1 (by averaging STRAIGHT parameters across the four instances of the same Syl2 syllable within a set and splicing the resulting syllable onto Syl1). To ensure that all syllables sounded natural in these contexts, we constrained word selection such that all Syl2 syllables within a set started with the same speech segment. In this way, we could cross-splice syllables without conflicting co-articulatory information at the onset of Syl2. As a result, the point at which words and pseudowords diverged (the divergence point) occurred not at the onset of Syl2 but one speech segment later (post divergence point speech segments highlighted as bold in Fig. 1A). Additional stimuli were derived by replacing the Syl1, Syl2, or both

syllables, with spectrally and intensity (RMS) matched noise (data for these stimuli are not reported here). In total there were eight sets of items and therefore 240 stimuli (32 items in each of the following conditions: Strong + Match, Weak + Match, Strong + Mismatch, Weak + Mismatch; 32 each of Strong + Noise, Weak + Noise; 32 of Noise + Syl2; 16 of Noise + Noise). For analysis of stimulus properties, see Supplementary Information.

Before the experiment, participants completed a brief practice session lasting approximately 3 min that contained examples of all the stimulus types included (spoken words, pseudowords with and without noise) but using different items to the main experiment. Stimulus delivery was controlled with Psychtoolbox 3.0.14[73] scripts in Matlab.

### Procedure (during MEG and fMRI recordings)

Stimulus presentation and task procedure was nearly identical for MEG and fMRI listeners. On each trial, listeners heard one of the spoken items separated by a stimulus onset asynchrony (SOA) of 2.5 sec (see Fig. S1A). For the MEG experiment, this SOA was jittered by adding a random time interval of ±100 ms. For the fMRI experiment, a constant SOA was used, and stimuli were presented at the offset of the scans (sparse imaging sequence). Participants were asked to look at a fixation cross in the centre of the screen and respond as quickly and accurately as possible whenever they heard a brief pause in the speech stimuli. Pauses consisted of an additional 200 ms silence inserted between Syl1 and Syl2. Pilot testing revealed that pause detection was more difficult when one or both syllables were replaced by noise.

Trials were randomly ordered during each of five presentation blocks of 288 trials. Each block consisted of a single presentation of all stimuli plus 'Pause' stimuli. Across all five blocks, the order of 'Pause' and 'No Pause' items was completely randomised; in total there was one 'Pause' item for every six 'No Pause' items, with this ratio constant across stimulus type (Strong + Match, Weak + Match etc.). Participants received feedback on performance (% correct) at the end of each block.

**Cued recall task (following fMRI scans).** After the fMRI scans, listeners completed a cued recall test assessing memory for the pseudowords. Data were not available for this task for four participants either because of technical issues or because the participant did not understand the task instructions. The final sample size consisted of 17 participants for this test. We cued participants with the first syllable of each pseudoword item by presenting the same stimuli as the MEG and fMRI experiments but replacing the second syllables with spectrally and intensity (RMS) matched noise. We refer to these as Strong+Noise and Weak+Noise items. Participants were told that the nonsense (pseudo) words were partly hidden by noise and were asked to recall the nonsense word by saying it aloud. Responses were scored for accuracy based on counting the percentage of trials in which participants generated the correct Syl2 used in the Mismatch items. The order of stimulus presentation was randomised for each participant.

**Web-based gating study.** Our measure of p(Syl2|Syl1) is derived from frequency counts and phonological transcriptions in the CELEX database. To validate this measure, we conducted a web-based gating task using the Strong+Noise and Weak+Noise stimuli (each item presented once in a random order). Participants were British English speakers (N = 110, recruited via https://www.prolific.co and non-overlapping with participants in the MEG/fMRI studies), aged 18 to 40 years and self-reported as having no hearing difficulties or language-related disorders. They were told that they would hear real English words partly hidden by noise and asked to report the identity of the word by typing it out. For each item, we calculated the proportion of participants reporting the source word, i.e., the matching word prior to replacing Syl2 with noise.

## Computational simulations

Both sharpened signal and prediction error computations have the goal of implementing Bayes perceptual inference for spoken words[2,9]:

$$p\left(Word_i | Evidence\right) = \frac{p\left(Word_i\right) \times p\left(Evidence | Word_i\right)}{\sum_{j=1}^{j=n} p\left(Word_j\right) \times p\left(Evidence | Word_j\right)} \quad (2)$$

This equation computes the conditional (posterior) probability of each word, given their prior probabilities (frequency of occurrence) and the available sensory evidence. We modelled uncertainty in the sensory evidence based on the acoustic similarity between individual speech segments. Acoustic representations were obtained from a large set of 1030 words spoken by the same speaker as for the present stimuli. These stimuli comprised recordings from previous studies[74–76] as well as the 32 Strong Match items and 32 Weak Match items from the current study. From these stimuli, we computed acoustic energy for individual speech segments (phonemes): 47 unique segments in total, averaged over 3659 repeated tokens (median token count = 52). Note that model representations consisted of 48 segments in total because of an additional unit to represent syllable boundaries (see below). Acoustic energy was expressed as a function of time, frequency and spectrotemporal modulations[77], as computed by the nsltools toolbox in Matlab (http://nsl.isr.umd.edu/downloads.html; for details see ref. 21). Segment onset times were obtained using a forced-alignment algorithm included as part of BAS software[78]; https://clarin.phonetik.uni-muenchen.de/BASWebServices/interface/WebMAUSGeneral. Acoustic representations were vectorized and averaged over multiple productions of the same segment. We then obtained a between-segment similarity matrix using negative Euclidean distances and normalised each row of this matrix using the softmax function (combined with a temperature parameter $T$) such that each row could be interpreted as segment probabilities:

$$p(Segment_i) = \frac{\exp(-dT)}{\sum_{i=1}^{i=n} \exp(-dT)} \quad (3)$$

where d denotes the Euclidean distance. From these segment probabilities, we could compute the sensory evidence for a word by applying the product rule over word segments (see ref. 2). The sensory evidence for an unfolding word will be strong if its constituent segments match internal representations that are acoustically dissimilar to other speech sounds in the language. It will also be strong if there are few word competitors sharing the same speech segments. Given that our stimulus set consisted of clear speech recordings, a reasonable assumption is that individual segment probabilities are close to one (indicating high certainty). Nonetheless, even clear speech segments will be perceived with momentary ambiguities as sub-phonemic information unfolds over the duration of a segment. In the absence of human sub-phonemic confusion data[79] for the British accent of the present stimuli, we scaled acoustic similarities to simulate a range of overall sensory uncertainty levels by varying the Softmax temperature parameter. The same qualitative patterns were present over a range of sensory uncertainty levels (Fig. S3). The one exception to this is apparent for simulations with the highest degree of sensory uncertainty for which prediction error representations in the Match condition are enhanced relative to simulations with reduced sensory uncertainty (compare the height of the first bar in each of the bottom row of plots in Fig. S3B). The effect of this is that—under high uncertainty—results show a reverse pattern with enhanced neural representations for Strong versus Weak items. This is consistent with previous modelling and experimental data[18,21] and is explained by an increase in the informativeness of prediction errors that occurs when strong predictions match noisy or ambiguous sensory input: Even though speech segments are accurately predicted, their acoustic form is not as expected (deviating from clear speech).

Note that our stimuli were selected based on syllable conditional probabilities p(Syl2|Syl1), which used the syllabified phonetic transcriptions in CELEX. For consistency, we therefore incorporated the syllable boundaries from CELEX into our model simulations. We achieved this by adding a separate segment to the model's inventory of segments, representing the boundary between syllables. Thus, an item like "bingo" was represented by six segments in total (five speech sounds plus the boundary between the two component syllables). In Fig. 1B, C, we omit the syllable boundary segment for display purposes.

To simulate speech processing as each speech segment in our stimuli was heard, we used Bayes theorem to compute the posterior probability distribution over all words in the lexicon. Then to compute predictions for each upcoming speech segment, we summed posterior probabilities over words sharing the same segment. These segment predictions were then multiplied by individual segment probabilities (derived from acoustic similarity as explained above), to simulate the sharpened signal computation (normalising to sum to one). For simulating the prediction error computation, segment predictions were instead subtracted from segment probabilities. For analysis of the simulated representations, pattern distances were computed as the Euclidean distance between model signals for pairs of items at the divergence point. For the signal magnitude analysis, we analysed model representations specifically for the speech segment that was heard at the divergence point.

## MEG acquisition and analysis

Magnetic fields were recorded with a VectorView system (Elekta Neuromag, Helsinki, Finland) containing two orthogonal planar gradiometers at each of 102 positions within a hemispheric array. Data were also acquired by magnetometer and EEG sensors. However, only data from the planar gradiometers were analysed as these sensors have maximum sensitivity to cortical sources directly under them and are therefore less sensitive to noise artifacts[80]. To monitor eye and heart activity, EOG and ECG signals were recorded with bipolar electrodes.

MEG data were processed using the temporal extension of Signal Source Separation[81] in Maxfilter 2.2 software to suppress noise sources, compensate for motion, and reconstruct any bad sensors. Also using MaxFilter, the data were transformed to a common coordinate frame for each participant separately (to compensate for changes in head position between blocks). Subsequent processing was done in SPM12 r7487 (Wellcome Trust Centre for Neuroimaging, London, UK), FieldTrip as distributed within SPM12 (Donders Institute for Brain, Cognition and Behaviour, Radboud University Nijmegen, the Netherlands) and The Decoding Toolbox v3.991[82] implemented in Matlab.

MEG data were downsampled to 250 Hz and ICA used to remove eye and heart artefacts (based on an automatic procedure correlating the component time-series with EOG and ECG signals). When epoching, we retained only the no pause trials. MEG recordings were epoched from −1700 ms to 2000 ms relative to the onset of Syl2 and epochs with large amplitude signals were discarded (time- and channel-averaged power larger than 2 standard deviations from the condition-specific mean for signal magnitude analysis; power larger than 3 standard deviations from the pooled mean for pattern analysis). For signal magnitude (evoked response) analyses only, the data were baseline-corrected from −750 to −500 ms relative to Syl2 onset; this corresponds to the silence period before speech onset (since the duration of Syl1 ranged from 130 to 479 ms; see Fig. S4C for distribution of Syl1 duration).

For time-frequency analysis, we computed the power between 2 and 48 Hz (in 1 Hz steps) by convolving the data from each trial with Morlet wavelets of 7 cycles. To analyse high frequency activity (52–90 Hz), we used multi-taper estimation (±10 Hz smoothing, 200 ms time-windows with a step size of 20 ms). We analysed differences in power attributable to our experimental manipulations without baseline normalisation. For time-domain analyses, the data were

additionally low-pass filtered below 30 Hz prior to trial averaging. Note that in the case of signal magnitude analyses, MEG responses reflect the average of 160 trials before outlier removal (5 repetitions × 32 items per stimulus type) while for pattern analyses, MEG responses reflect the average of 5 repetitions (since here we analyse item-specific responses). Subsequent analysis was based on the period immediately before and after Syl2 onset (−200 to 1000 ms).

For signal magnitude (evoked response) analysis, the MEG data across the sensor array were summarised as the root mean square (RMS) over the 20 sensors with the largest evoked response (separately in each participant and hemisphere). Essentially the same results are obtained when selecting 10 or 40 sensors. The same sensor selections used for the signal magnitude (evoked response) analysis were also used when analysing time-frequency power. For pattern analyses, neural pattern distances were computed by taking the Euclidean distance between sensor patterns (over the entire 204 sensor-array) evoked by each pair of items in the relevant stimulus set. To assess whether MEG responses reflected the phonetic information about the syllables, neural pattern distances were correlated with the phonetic dissimilarities between the syllables of item pairs using Fisher-transformed Spearman's correlations. To compute the phonetic dissimilarities we used the Levenshtein distance between the phonetic transcriptions for each syllable, normalised by the maximum number of segments within the two syllables being compared; shown in Fig. 4A for Syl2 and Fig. S5 for Syl1 (see ref. [77] for a similar method applied to scoring phonetic responses). To maximise statistical power for this particular analysis, correlations were computed both across- and within-condition (see Fig. 4A). As shown in Fig. 4A, we also excluded dissimilarity of item pairs such as "cargar" and "cougo" since similar but opposite prediction errors are evoked during Syl2 of these items. For example, for "cargar", listeners predict /g @U/ but hear /g @ r*/. For "cougo", listeners predict /g @ r*/ but hear /g @U/. The pattern of prediction error in these two cases is identical, albeit opposite in polarity. Given uncertainty on how the polarity of prediction errors is encoded in neural responses[83], we exclude these pseudoword item pairs from pattern analysis of our neural data and model simulations. We also exclude the corresponding word item pairs (e.g., "cougar" and "cargo"; shown as NaNs in Fig. 4A).

When correlating syllable conditional probabilities with MEG responses (either magnitude-based or pattern-based measures), we also used Fisher-transformed Spearman's correlations. For pattern analysis, in which the dependent measures reflect pattern distances for pairs of items, we averaged syllable conditional probabilities for the items in each pair of words (after log transformation).

For both signal magnitude (RMS signals, time-frequency power or Spearman correlations) and pattern analyses (pattern distances or Spearman correlations), group-level one-sample $t$-tests were performed for each timepoint (and frequency bin when relevant) while controlling the family-wise error (FWE) rate using a non-parametric cluster-based permutation procedure based on 5000 iterations[84]. These tests were performed on contrasts of the data, testing for the main effects and interactions in our experimental design, as well as any simple effects. Reported effects were obtained by using a cluster defining height threshold of $p < 0.05$ with a cluster $t$-sum threshold of $p < 0.05$ (FWE corrected), unless otherwise stated.

## fMRI acquisition and analysis

Imaging data were collected on a Siemens 3 Tesla Prisma MRI scanner (http://www.siemens.com). A total of 291 echo planar imaging (EPI) volumes were acquired in each of 5 scanning runs, using a 32-channel head coil and a multiband sparse imaging sequence (TR = 2.50 s; TA = 1.135 s; TE = 30 ms; 48 slices covering the whole brain; flip angle = 78 deg; in-plane resolution = 3 × 3 mm; matrix size = 64 × 64; echo spacing = 0.5 ms; inter-slice gap = 25%). After the third run, field maps were

acquired (short TE = 10 ms; long TE = 12.46 ms). The experimental session commenced with the acquisition of a high-resolution T1-weighted structural MRI scan (TR = 2250 ms; TE = 2.99 ms; flip angle = 9°; 1 mm isotropic resolution; matrix size: 256 × 240 × 192 mm; GRAPPA acceleration factor PE = 2; Reference lines PE = 24).

fMRI pre-processing was performed in SPM12 r7219. After discarding the first four volumes to allow for magnetic saturation effects, the remaining images were realigned and unwarped to the first volume to correct for movement of participants during scanning. Also at the unwarping stage, the acquired field maps were used to correct for geometric distortions in the EPI due to magnetic field variations. Realigned images were co-registered to the mean functional image and then subjected to statistical analysis. For signal magnitude analysis, prior to further processing, images were also normalised to the Montreal Neurological Institute (MNI) template image using the parameters from the segmentation of the structural image (resampled resolution: 2 × 2 × 2 mm) and smoothed with a Gaussian kernel of 6 mm full-width at half-maximum. For pattern analysis, normalisation and smoothing were performed only after computing pattern distances.

Statistical analysis was based on the general linear model (GLM) of each participant's fMRI time series, using a 1/128 s highpass filter and AR1 correction for auto-correlation. The design matrix comprised the auditory stimulus events (onset of the spoken words), each modelled as a stick (delta) function and convolved with the canonical haemodynamic response function specified in SPM software. For signal magnitude analysis, separate columns were specified for each of the stimulus types in addition to a column for button presses (replicated for each of the five experimental blocks). For pattern analysis, the design matrix was identical but separate columns were specified for each individual item.

Pattern analysis was performed using searchlight analysis[85] as implemented in the Decoding toolbox (v3.991), using spheres with a radius of 8 mm and constrained to voxels within the whole-brain mask generated by SPM during model estimation. Pattern distances were computed using the cross-validated Mahalanobis distance[28]. This 'Crossnobis' distance is equivalent to the (cross-validated) Euclidean distance, normalised by the noise covariance between voxels (estimated from the GLM residuals). Cross-validation ensures that the expected pattern distance is zero if two voxel patterns are not statistically different from each other, making it a readily interpretable distance measure. Note that because of this cross-validation, the Crossnobis distance can sometimes be negative if it's true value is close to zero in the presence of noise. The Crossnobis distance has been demonstrated to be a more reliable and accurate metric for characterising multivoxel patterns than the correlation or Euclidean distance[28]. To assess whether fMRI responses reflected the phonetic information about Syl2, neural pattern distances were Spearman correlated with the phonetic dissimilarities between Syl2 portions of item pairs using the same Levenshtein distance measure as the MEG analysis (shown in Fig. 4A).

For both signal magnitude (contrast images from first-level models or Spearman correlations) and pattern analyses (Spearman correlations), group-level one-sample $t$-tests were performed for each voxel within a mask covering superior and middle temporal regions (shown in Fig. 3A). Inference was conducted parametrically, controlling the family-wise error (FWE) rate using random field theory. Reported effects were obtained by using a cluster defining height threshold of $p < 0.001$ with a cluster extent threshold of $p < 0.05$ (FWE corrected), unless otherwise stated.

## Reporting summary

Further information on research design is available in the Nature Portfolio Reporting Summary linked to this article.

## Data availability

The stimuli and data for this study are available on OSF (https://osf.io/wjd4s/). Source data are provided with this paper.

## Code availability

The code for the analyses and computational simulations in this study are available on OSF (https://osf.io/wjd4s/).

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

## Acknowledgements

This research was supported by intermural funding from the Medical Research Council to the Cognition and Brain Sciences Unit (MC_UU_0005/5 and MC_UU_00030/6 to M.H.D.). We thank Sander Van Bree and Clare Cook for assistance with MEG data collection and Steve Eldridge and Karen Kabakulu for assistance with radiography.

## Author contributions

E.S. and M.H.D. designed the study; E.S. and M.H.D. developed the stimuli; E.S. and L.B. collected the data; E.S. performed the analysis; E.S. developed the computational models; All authors contributed to the development of the analytic approach and the interpretation of the results; E.S. wrote the first draft; All authors contributed to the final manuscript.

## Competing interests

The authors declare no competing interests.
