## [Transparent Peer Review file · Nature Communications]

Convergent neural signatures of speech prediction error are a biological marker for spoken word recognition

Corresponding Author: Dr Ediz Sohoglu

Version 0:

Reviewer comments:

Reviewer #1

(Remarks to the Author)

In this study, the researchers used both MEG and fMRI together with computational simulations, to investigate how the brain combines predictions with incoming speech input in the superior temporal cortex. They manipulated the predictive strength of the first syllable on the second syllable (strong vs. weak) and the compatibility between the second syllable and the first syllable (match vs. mismatch vs. noise). They used both univariate amplitude and multivariate pattern analyses to explore the neural representation of speech, particularly focusing on the mechanisms of representational sharpening and prediction error. The authors present evidence supporting the prediction error mechanism through various analyses, including greater univariate responses (both evoked and oscillatory activity) and enhanced representations of second syllables that violated the constraints of the first syllable or followed weakly constraining first syllables. They also included computational simulations to support their conclusions. While the manuscript is clearly presented, I have a few concerns regarding the underlying assumptions of the hypotheses and the multivariate results.

1, I have some reservations about the distinction between representational sharpening and prediction error as discussed in this study. As the authors themselves acknowledge in the introduction, there's existing literature suggesting that these two mechanisms might not be mutually exclusive, with some proposals suggesting they operate in parallel and localize to different cortical layers.

Previous research on representational sharpening has primarily focused on detecting neural representations of expected inputs, even when they elicit relatively small overall neural activity. Typically, this is done by comparing the decodability of expected and unexpected inputs. Better decoding performance for expected inputs is often interpreted as evidence for representational sharpening of the expected information. However, it's important to note that the sharpening mechanism doesn't necessarily imply that unexpected information is not represented when prior predictions are disconfirmed. If unexpected bottom-up inputs are still represented, they should also be decodable. Therefore, achieving better decoding performance for unexpected inputs doesn't inherently mean that expected inputs are not represented or that their representation isn't sharpened.

In the context of this current study, the authors argue that a better neural representation of unexpected inputs can provide evidence for the representation of prediction error. Although I agree that this could support the idea of a representation of prediction error, I don't think it necessarily implies that expected inputs are not represented.

In summary, I believe the contrast between sharpening and prediction error mechanisms is problematic in this study. Therefore, the findings may be more appropriately used to support the concept of neural representation of prediction error rather than making definitive claims about the absence of representation for expected inputs.

2, I have a few questions regarding the validity of the multivariate analyses in this study. The authors employed a method where they compared the neural representation between different conditions by measuring the distance of the neural activity produced by the inputs. They interpreted a greater neural pattern distance produced by unexpected inputs as indicating a better representation of those inputs. However, there are several concerns with this approach.

First, the neural pattern distance should depend on the specific relationship between the evoking stimuli. For instance, when the stimuli are relatively similar to each other, a smaller neural distance would actually reflect a better neural representation, as similar stimuli should produce more similar neural patterns. To address this issue, it will be more appropriate for the authors to correlate the neural dissimilarity matrix with the phonetic dissimilarity matrix separately for each condition (as what they did for all stimuli) and then compare the correlation values between different conditions.

Second, the use of Euclidean distance to measure neural pattern distance in the MEG data is problematic. Euclidean

distance is sensitive to the amplitude of signals, meaning that larger univariate signals will lead to greater neural pattern distance values. Consequently, the similar univariate and multivariate effects observed in MEG data could potentially be driven by this confounding factor. It is also sensitive to the amplitude of noise, meaning that greater noise would also lead to larger Euclidean distance. Therefore, the greater neural pattern distance could also be explained by larger noise in the data. To mitigate these concerns, it would be advisable for the authors to employ cross-validated Euclidean distance, similar to what they did for the fMRI data, which would help remove this potential confound and provide more reliable results.

3, While the authors argue for the convergence of MEG and fMRI findings, the fMRI univariate effect seems to be more consistent with the MEG effect observed in a later time window, specifically beyond 400ms. This finding is consistent with previous research indicating that fMRI may be less sensitive to early transient MEG activity. This raises questions about the localization of the MEG effects within the brain regions identified by fMRI. One approach is to conduct both univariate and multivariate analyses on the MEG source-localized data. This would allow for a more detailed exploration of the timing and localization of the observed effects, potentially providing a clearer picture of the relationship between MEG and fMRI findings.

4, It would be helpful if the authors could provide a specific example illustrating the calculation of word probabilities containing the target segment. It is hard to know how the between-segment similarity matrix is used incrementally for every segment within a word.

Reviewer #2

(Remarks to the Author)

The authors present an MEG and fMRI experiment to differentiate two theories of expectation-driven (speech) perception. Unlike their earlier work, which used written cues and heavy speech distortion to induce expectations, this study uses syllabic conditional probabilities within words to represent prior knowledge. It investigates the interaction between prediction strength and “prediction (mis)match”. Only the prediction error model predicts such an interactive effect. Evidence for such an interaction is found in both MEG and fMRI data.

This is an interesting study that might be potentially of wide interest. I appreciate the clever design, which is an advance over prior studies from the same authors that used written cues. However, I have a number of questions that need to be addressed before I fully appreciate the result.

Major comments/questions:

- Since the authors cast prediction in probabilistic terms – as $P(\text{Syl2} | \text{Syl1})$ – I had a hard time making sense of “prediction strength” and “mismatch” as independent factors that can interact. To my mind, once you think of prediction probabilistically (rather than categorically), isn't any “mismatch” just an item with low probability? Treating M and MM as independent splits, and test for an interaction, then becomes splitting up a continuum and performing independent test of the same underlying effect (unexpectedness), once at the extreme of unexpectedness (among “mismatches”, which in a proper probabilistic model would have small but different, non-zero probabilities), and another in a wider chunk of the distribution (among “matches”). Since this interaction is the key conceptual innovation, I really need some help here, I currently find it very difficult to make sense of. It seems very different from the interactions between independent factors (sensory noise and prediction strength) that the authors reported in prior research in this tradition.

- I was struck by the fact that the “sharpened signals” model fails to recapitulate unexpectedness effects or “expectation suppression” (magnitude of $MM < M$ in both prediction strength “conditions”). The authors stress that the pattern simulations are more important, and that magnitude simulations “ignore contributions from neurons tuned away from the stimulus”, which they call a “questionable simplification” (l. 285-291). Yet, I was puzzled by this pattern. It seems to make the experiment unnecessary as it allows us to rule out the pure “sharpened signal” model a priori (since predictability effects are well-known to exist, even without noise, as demonstrated e.g. by the same senior author in Gagnepain et al. 2012). Does this limitation apply to earlier work from partly the same authors? Do those simulations equally predict that the sharpened signals model predicts no expectation suppression (but expectation enhancement) when stimuli are presented without noise? If so, doesn't that undermine the entire endeavor of using this “sharpened signal” model for simulations?

- While I really like the idea of the result in Figure 6 (decoding signal from noise or “restored syllables”, and finding an interaction, which could only be explained by prediction errors), I find the result hard to interpret. No statistics are reported other than $P < 0.05$ (which suggests $0.01 < p < 0.05$, but is quite ambiguous). No error bars are visualized. And as far as I can see there is also no statistical evidence for a main effect of “above chance” decoding in the first place. This makes it very difficult to me to assess the strength of this evidence.

- Figure 3: Isn't it a problem that the main correlations in fMRI signal magnitude and $P(\text{Syl2} | \text{Syl1})$ are positive? The authors write that this replicates the pre-onset effects in Figure 2, but when describing these results they urge caution because these may well reflect acoustic differences in Syl1 (line 343-347).

- The authors do not report any error bars on any of the time courses. It would be helpful and important to visualize not just means but also consistencies/variability.

Minor points / thoughts:

- A schematic of the modelling steps (extraction of segments, acoustic similarities to get likelihoods, use of lexical frequencies to get prior) would be very helpful because it is currently quite tricky to follow all details, and these aren't standard probabilities (as standard word surprisal from a language model might be).
- Line 315-320 – why select 20 channels? Surely the results are not fully dependent on this arbitrary number? Can the authors demonstrate some robustness?

- Phonetic RDMs are based on Levenshtein distances between phonetic descriptions. Figure 4B suggests that the RDM still contains “enough” information, but it seems like quite the simplification. Why not use acoustic distances as the authors use in their simulations? Or is that very different? (Just a clarification question).

Reviewer #3

(Remarks to the Author)

Let me start by saying that although I am reasonably fluent in EEG, which carries over to MEG somewhat, I have not carried out an MEG or fMRI study, so I cannot comment on the soundness of the use of these tools.

This study builds on the authors' work exploring the neural coding of speech, with a focus on whether anticipatory processing is most accurately characterized as sharpening an expected signal or as reducing prediction error. The two alternatives are difficult to distinguish because they often lead to the same perceptual outcome (data pattern). Across two experiments, the data provide the most compelling case for prediction error to date. Further, they provide insight into its time course and the brain regions involved. This is an important study. Suggestions below are intended to increase the impact of the work and improve clarity.

As written, the manuscript is very much an empirical study that adjudicates two hypothesized mechanisms. What are the theoretical implications of the results? The authors touch on these a bit, but I think far more could be done to link the findings to theory. It has been a challenge to link neural data with other mechanistic models of word processing in a substantive way. The current data can begin to place truly neurally plausible constraints on architecture. How might local suppression be implemented and how does it interface with other mechanisms involved in recognition (e.g., syllable 1 activation). A time-course account over the two-syllable stimuli would be welcome. Discussion of how this account extends beyond words to sentences and discourse would help expand the explanatory significance of the prediction error account. Ultimately we want an account that spans many levels of analysis. These data have the exciting potential to start bridging levels.

Part of achieving the above could include making the model more prominent. It is the source of the predictions and the cornerstone of the study, yet its presentation is relegated to the Method section in a handful of paragraphs. If we are to take it seriously, integrate it into the discussion and expand its description. If the authors are not committed to the implementation (e.g., they view it as toy model or no more than a proof of concept), then this position should be stated. I view this as one of the strengths of the paper, which appears on their past work.

Place the noise data in a separate manuscript, maybe one in which snr is manipulated and modeled. The data and their lengthy discussion are intriguing, but in contrast to the no-noise data, their interpretation is much more speculative and somewhat off topic. They are not needed to support the authors' case.

The terse writing in the Method section is frustrating when trying to understand what was done or how it was done. As an example, only by referring to Figure 1 was I able to understand stimulus construction across conditions. The modeling paragraphs are challenging as well.

Add captions to the figures in the supplement.

Figure 1 is extremely helpful (the figures are very clear overall), but the illustrative example of "bingo" in the text and figure is confusing. The first syllable should be "BIHNG" not "BIHN" (ARPABET spelling), right?

-Mark (signed)

Version 1:

Reviewer comments:

Reviewer #1

(Remarks to the Author)

Thank you for addressing my questions. I appreciate your efforts in clarifying the points I raised in the previous manuscript.

Reviewer #2

(Remarks to the Author)

It was a bit hard to review all the revisions because the changes were not tracked in the PDF.

I would like to thank the authors for the effort, my comments and questions have been sufficiently addressed.

I do have some conceptual reservations regarding the flexibility of the simulations with the choice of a linking function, and what this means for the strength of the argument of the simulations in this paper and in prior papers from the same group, but I don't see it as a deal breaker for publication.

I also agree with Reviewer 1 about the distinction between prediction error and sharpened signals. The authors rebut that some models (like classic interactive activation models) only predict sharpening to occur. However, if I understand correctly,

the opposite does not seem to be true: most cortical predictive coding theories would predict both sharpening and 'prediction error' to occur. This potentially undermines the distinction the authors are after. I leave it up to R1 what to make of their response and revisions.

Reviewer #3

(Remarks to the Author)

The authors were responsive to my feedback. The revision is now a more tightly focused study. The amount of data presented is still large but now manageable. Although the writing is dense, it is clear and the authors do a good job of making the study accessible to a larger readership.

What makes this paper such a nice piece of work is the combination of the modeling and the experiments. By elevating the model in the revision in multiple places, their synergy is now evident, and the study takes on added theoretical significance. The model makes explicit pseudoword predictions, and only by analyzing its performance can one understand the cause of the unintuitive predictions of the predictive error account. The empirical data support these predictions across multiple, complementary analyses. The field of cognitive modeling is replete with examples just like this, and they consistently reinforce the value of modeling in providing insight that leads to substantive advances.

Acknowledgement of the limitations of the model and the use of pseudowords (they surely engage or mechanisms) is welcome.

One oddity to consider addressing before publication (not a requirement): Frequentist statistics are used to support inferences in various places. In light of the sophisticated acoustic modeling, stimulus construction and data processing techniques throughout the manuscript, this is incongruous. The authors are candid about weaker effects; p-values are a less-than-ideal means of conveying their strength.

We thank the reviewers' positive and constructive comments on the previous version of our manuscript. In the revised version, we have made significant changes that we hope addresses all of these comments.

Below we include excerpts from the decision letter (in **bold**), along with our detailed response (in plain text) and sections of text from the revised paper (in *italics*).

Reviewer #1:

In this study, the researchers used both MEG and fMRI together with computational simulations, to investigate how the brain combines predictions with incoming speech input in the superior temporal cortex. They manipulated the predictive strength of the first syllable on the second syllable (strong vs. weak) and the compatibility between the second syllable and the first syllable (match vs. mismatch vs. noise). They used both univariate amplitude and multivariate pattern analyses to explore the neural representation of speech, particularly focusing on the mechanisms of representational sharpening and prediction error. The authors present evidence supporting the prediction error mechanism through various analyses, including greater univariate responses (both evoked and oscillatory activity) and enhanced representations of second syllables that violated the constraints of the first syllable or followed weakly constraining first syllables. They also included computational simulations to support their conclusions. While the manuscript is clearly presented, I have a few concerns regarding the underlying assumptions of the hypotheses and the multivariate results.

1. I have some reservations about the distinction between representational sharpening and prediction error as discussed in this study. As the authors themselves acknowledge in the introduction, there's existing literature suggesting that these two mechanisms might not be mutually exclusive, with some proposals suggesting they operate in parallel and localize to different cortical layers.

Previous research on representational sharpening has primarily focused on detecting neural representations of expected inputs, even when they elicit relatively small overall neural activity. Typically, this is done by comparing the decodability of expected and unexpected inputs. Better decoding performance for expected inputs is often interpreted as evidence for representational sharpening of the expected information. However, it's important to note that the sharpening mechanism doesn't necessarily imply that unexpected information is not represented when prior predictions are disconfirmed. If unexpected bottom-up inputs are still represented, they should also be decodable. Therefore, achieving better decoding performance for unexpected inputs doesn't inherently mean that expected inputs are not represented or that their representation isn't sharpened.

In the context of this current study, the authors argue that a better neural representation of unexpected inputs can provide evidence for the representation of prediction error.

Although I agree that this could support the idea of a representation of prediction error, I don't think it necessarily implies that expected inputs are not represented.

In summary, I believe the contrast between sharpening and prediction error mechanisms is problematic in this study. Therefore, the findings may be more appropriately used to

support the concept of neural representation of prediction error rather than making definitive claims about the absence of representation for expected inputs.

We agree it is plausible that sharpened signal and prediction error representations co-exist, possibly in distinct neural populations (as per some formulations of predictive coding e.g. Rao & Ballard 1999 *Nat Neuro*). However, we maintain it is informative to contrast these two representational schemes as distinct accounts. This is because influential models such as TRACE propose only sharpened signal representations. Thus, our evidence that cortical responses to speech are best explained by prediction error representations clearly challenge models like TRACE and instead support predictive coding models.

We now have added text in the Discussion section to acknowledge and elaborate on this point (page 26, line 693):

It should be noted that in prediction error accounts, computation of prediction errors is not the end goal of the system [as discussed in ref 21]. Rather, they are used to update – and thereby enhance or sharpen – future predictions. In some formulations of predictive coding [8,17], these sharpened predictions are represented in separate neural populations. However, we saw little evidence of sharpened representations in our study; perhaps because these sharpened predictions reside in deeper cortical layers [17] to which MEG [31] and BOLD [32] signals are less sensitive. It has also been proposed that prediction errors and predictions are communicated via distinct oscillatory channels [33,34], although our time-frequency analysis did not provide support for this (discussed below). In future, methodological advances e.g. in layer-specific imaging [35], or more invasive methods [36] may help to resolve this issue. Insights may also come from studies that simultaneously probe multiple levels of processing (e.g. to observe prediction updating in higher-level semantic stages of processing).

2. I have a few questions regarding the validity of the multivariate analyses in this study. The authors employed a method where they compared the neural representation between different conditions by measuring the distance of the neural activity produced by the inputs. They interpreted a greater neural pattern distance produced by unexpected inputs as indicating a better representation of those inputs. However, there are several concerns with this approach.

First, the neural pattern distance should depend on the specific relationship between the evoking stimuli. For instance, when the stimuli are relatively similar to each other, a smaller neural distance would actually reflect a better neural representation, as similar stimuli should produce more similar neural patterns. To address this issue, it will be more appropriate for the authors to correlate the neural dissimilarity matrix with the phonetic dissimilarity matrix separately for each condition (as what they did for all stimuli) and then compare the correlation values between different conditions.

As our computational simulations in Figure 1E of the manuscript show (which follow the same analysis approach as for our MEG analysis), condition-wise differences in the size of pattern distances are sufficient to distinguish sharpened signal from prediction error accounts. We therefore argue that the reported analysis of MEG pattern distances is valid.

As well as being sufficient to distinguish accounts, this approach has the advantage of not requiring any assumptions as to the precise form of representations that are modulated by our experimental manipulations (e.g. phonetic versus acoustic, segments versus syllables etc.).

Nonetheless, we agree that it would be helpful to supplement this analysis with the one suggested by the reviewer. This analysis is now reported in the Results section (page 20, line 546) and in Figure S2D of the manuscript. The results remain broadly consistent with the previously reported analysis based on the size of pattern distances. While we did not observe an interaction between prediction strength and congruency, as before we did observe a positive effect of prediction strength within the Mismatch condition. This result favours the prediction error account.

We thank the reviewer for this suggestion, which we think has strengthened our manuscript.

Second, the use of Euclidean distance to measure neural pattern distance in the MEG data is problematic. Euclidean distance is sensitive to the amplitude of signals, meaning that larger univariate signals will lead to greater neural pattern distance values. Consequently, the similar univariate and multivariate effects observed in MEG data could potentially be driven by this confounding factor. It is also sensitive to the amplitude of noise, meaning that greater noise would also lead to larger Euclidean distance. Therefore, the greater neural pattern distance could also be explained by larger noise in the data. To mitigate these concerns, it would be advisable for the authors to employ cross-validated Euclidean distance, similar to what they did for the fMRI data, which would help remove this potential confound and provide more reliable results.

As per the reviewer's suggestion, we have repeated this analysis based on cross-validated Euclidean distances with noise normalization (i.e. cross-validated Mahalanobis or 'Crossnobis' distances; shown in Figure R1 below). We use this analysis method to assess whether the MEG signal contained phonetic information about the syllables i.e. the Crossnobis equivalent of Figure 4B previously reported which used Euclidean distances.

We find that this Crossnobis analysis results in qualitatively similar results as the analysis previously reported (based on standard Euclidean distances) although cross-subject reliability appears weaker (compare with Figure 4B in the manuscript). This reduced reliability may be the consequence of cross-validation: Although cross-validation affords benefits in terms of providing an unbiased measure of representational dissimilarity, this comes at the cost of increased variance (Diedrichsen et al. 2020 arXiv). Thus, cross-validation does not always increase reliability and sometimes can even reduce reliability, as we find here.

Given the outcome of this analysis, we think it is appropriate to state in the manuscript that we have performed the alternative analysis using Crossnobis distances and report that it produced qualitatively similar but less reliable results. This is now mentioned in the revised manuscript (Results section, page 20, line 533).

Figure R1- Group-averaged correlation between cross-validated Mahalanobis MEG pattern (Crossnobis) distances and phonetic dissimilarities for the second syllable (Syl2; in red) and first syllable (Syl1; in beige). Horizontal bars indicate timepoints when correlations were significantly different from zero (thick lines show corrected significance at $p < .05$ clusterwise; thin lines show uncorrected significance at $p < .05$).

3. While the authors argue for the convergence of MEG and fMRI findings, the fMRI univariate effect seems to be more consistent with the MEG effect observed in a later time window, specifically beyond 400ms. This finding is consistent with previous research indicating that fMRI may be less sensitive to early transience MEG activity. This raises questions about the localization of the MEG effects within the brain regions identified by fMRI. One approach is to conduct both univariate and multivariate analyses on the MEG source-localized data. This would allow for a more detailed exploration of the timing and localization of the observed effects, potentially providing a clearer picture of the relationship between MEG and fMRI findings.

We appreciate the suggestion to conduct our MEG analysis on source-localised data. This could provide further insights into univariate responses for which we observe multiple effects at different timepoints (an early main effect of Strong > Weak conditions and later interaction between prediction strength and congruency; see Figure 2A in manuscript for example). However, as we argue in the manuscript (echoing previous proposals; see Blank & Davis 2016 PLOS Biology; de Lange et al. 2018 Trends in Cognitive Sciences), it is difficult to distinguish sharpened signal versus prediction error accounts using univariate analysis. Therefore, we do not think the source-based univariate analysis the reviewer proposes will be informative with respect to the focus of our study.

The reviewer also proposes that the multivariate analysis is conducted in source space. However, we think this is problematic given the relatively low spatial resolution of MEG source-localized data (as all vertices within a spherical searchlight will be highly correlated). We also note that any source reconstruction method requires many assumptions to solve what is widely acknowledged to be an ill-posed problem (computation of the inverse solution).

We therefore think our sensor-space approach is valid and appropriate. This enables us to examine neural representations with millisecond accuracy in a relatively assumption free fashion. To gain insights into the underlying neural generators, we turn instead to our more spatially precise fMRI data. We see this combination of MEG and fMRI data as a major strength of our study.

4. It would be helpful if the authors could provide a specific example illustrating the calculation of word probabilities containing the target segment. It is hard to how the between-segment similarity matrix is used incrementally for every segment within a word.

We have now added a new supplementary figure (Figure S6) in the manuscript to provide a more detailed depiction of our computational simulations (see also our responses to related comments by Reviewers 2 and 3). Note that the between-item phonetic dissimilarity matrix is only used for neural analyses. For the calculation of word/segment probabilities in our computational simulations, we either take the sum of log-transformed probabilities (for univariate simulations) or average between-item pattern distance (for multivariate simulations). This is performed at a single timepoint corresponding to the divergence point of our stimuli.

Reviewer #2:

The authors present an MEG and fMRI experiment to differentiate two theories of expectation-driven (speech) perception. Unlike their earlier work, which used written cues and heavy speech distortion to induce expectations, this study uses syllabic conditional probabilities within words to represent prior knowledge. It investigates the interaction between prediction strength and “prediction (mis)match”. Only the prediction error model predicts such an interactive effect. Evidence for such an interaction is found in both MEG and fMRI data.

This is an interesting study that might be potentially of wide interest. I appreciate the clever design, which is an advance over prior studies from the same authors that used written cues. However, I have a number of questions that need to be addressed before I fully appreciate the result.

Major comments/questions:

1. Since the authors cast prediction in probabilistic terms – as $P(\text{Syl2} \mid \text{Syl1})$ – I had a hard time making sense of “prediction strength” and “mismatch” as independent factors that can interact. To my mind, once you think of prediction probabilistically (rather than categorically), isn’t any “mismatch” just an item with low probability? Treating M and MM as independent splits, and test for an interaction, then becomes splitting up a continuum and performing independent test of the same underlying effect (unexpectedness), once at the extreme of unexpectedness (among “mismatches”, which in a proper probabilistic model would have small but different, non-zero probabilities), and another in a wider chunk of the distribution (among “matches”). Since this interaction is the key conceptual innovation, I really need some help here, I currently find it very difficult to make sense of. It seems very different from the interactions between independent factors (sensory noise

and prediction strength) that the authors reported in prior research in this tradition.

We regret this aspect of our design was not sufficiently clear and we welcome the opportunity to provide clarity in our response here and through revisions of the manuscript.

The reviewer is correct to point out that if operationalising prediction strength as $p(\text{Syl2}|\text{Syl1})$, there should be little difference in expectedness between ‘Strong’ and ‘Weak’ items in the Mismatch condition i.e. all Mismatch items can be considered as being at the extreme (low probability) end of the unexpectedness continuum. However, in our study, Strong and Weak items differ not only in terms of $p(\text{Syl2}|\text{Syl1})$, but also in entropy over all possible syllables i.e. the overall uncertainty of predictions (see Figure 1A in manuscript). The distinction between $p(\text{Syl2}|\text{Syl1})$ and entropy echoes the distinction between cloze probability and sentence constraint that is made in the N400 literature (see Michealov & Bergen 2023 Cortex).

Therefore, if listeners make predictions probabilistically over distributions of speech sounds, we can still expect an effect of prediction strength for Mismatch items. Specifically, in the Weak+Mismatch condition, the unexpected syllable follows a state of imprecise (high entropy) predictions. By contrast in the Strong+Mismatch condition, the unexpected second syllable follows a state of precise (low entropy) predictions for different syllables (forming real words).

Importantly, this effect of Strong vs Weak for Mismatch items on speech representations is seen in our experimental data (e.g. MEG analysis in Figure 4D of manuscript), which would not be the case if our manipulations of prediction strength and match/mismatch reflected opposite ends of the same continuum based on $p(\text{Syl2}|\text{Syl1})$. Also critically, our simulations confirm that this effect distinguishes sharpened signal from prediction error representations (see Figure 1D in manuscript). Thus, we maintain that prediction strength and match/mismatch are distinct manipulations and are informative for distinguishing computational accounts.

To make this aspect of our study clearer, in the revised manuscript we have added text in the Results section when our manipulations are first described in detail (page 8, line 236):

It is important to note that even though $p(\text{Syl2}|\text{Syl1})$ approaches zero for all Mismatch items, we can still expect a difference in prediction strength between Strong+Mismatch and Weak+Mismatch items since Strong and Weak items differ not only in terms of $p(\text{Syl2}|\text{Syl1})$, but also in entropy over all possible second syllables i.e. the overall uncertainty of predictions (see SI Results and Figure S4A). Indeed in natural speech, conditional probabilities for upcoming speech sounds are negatively correlated with entropy [12,14,22]. Going back to the example above, after hearing the first syllable “bin”, entropy is relatively low because only one word (bingo) can be predicted. In this case, the unexpected second syllable (“gger”) follows a state of precise predictions (low uncertainty/entropy). For tango on the other hand, entropy is relatively high after hearing the first syllable because predictions are made for multiple second syllables each arising from different word candidates. Here the unexpected second syllable follows a state of imprecise predictions (high

uncertainty/entropy). Thus, while we constructed our stimuli based on $p(\text{Syl2}|\text{Syl1})$, our manipulation of prediction strength reflects differences in both $p(\text{Syl2}|\text{Syl1})$ and syllable entropy. Both aspects of predictability may contribute to perceptual and neural responses in the Match (real word) condition. In the Mismatch condition however, effects of prediction strength can only reflect differences in entropy (as explained above).

We also now provide discussion of how differences in entropy lead to the outcomes observed in our simulations (Discussion section, page 27, line 707):

Why are prediction error representations enhanced by prediction strength when predictions mismatch with speech input i.e. for pseudowords? Because the combinations of syllables in these items will never previously have been heard by listeners as English spoken words, the expected probabilities for the second syllables should always approach zero [in information theory terms, surprisal should be maximal; see ref 3]. Therefore, if neural representations were only sensitive to predictions for the heard syllables (and not alternative syllables), we would not expect any neural differences between Strong+Mismatch and Weak+Mismatch items. However, our manipulation of prediction strength reflects not only differences in predictions for the heard second syllables, but also in the uncertainty of predictions over all possible syllables: In information theory, this uncertainty is formulated as entropy [22]. Indeed in natural speech, predictions for specific speech sounds are negatively correlated with entropy [12,14,22].

We suggest our results reflect both aspects of prediction i.e. for the heard syllables as well as the uncertainty of predictions over all possible syllables. In the Strong+Mismatch condition, a focal (low entropy) prediction is misaligned with sensory input. As a result, there is increased information in neural prediction errors because the sensory signal remains “unexplained”. By contrast, weak and more distributed (lower entropy) predictions will tend to align more with speech input representations. This illustrated by the pseudoword tanger. After hearing the first syllable “tan”, predictions are spread weakly over multiple word candidates including the word tanker in which the penultimate sound is shared with tanger. This overlap results in reduced prediction errors in the Weak+Mismatch condition.

Our findings are consistent with neurophysiological findings showing that in addition to the informativeness of heard speech sounds, brain responses are also sensitive to the uncertainty of predictions made in parallel for alternative sounds [12,14,26,30]. Going beyond this previous work however, our current findings suggest that neural sensitivity to prediction uncertainty reflects the specific neural computation of prediction error.

Although we think the explanation above (based on prediction uncertainty) is a parsimonious explanation for our results, we cannot rule out the possibility that prediction violations for pseudowords may engage additional or distinct neural mechanisms to those supporting processing of familiar words. We have now added discussion of this point in the Discussion section (page 27, line 735):

Prediction uncertainty is a parsimonious explanation of the prediction strength effect on pseudoword representations. However, it is also possible that the strong violations of predictions that occur for pseudoword items may engage additional or qualitatively different mechanisms to those that support processing of familiar words. Bayesian inference for unfamiliar words is ill-defined because a pseudoword by definition has never previously been heard and therefore has a prior probability of zero [2,3]. In this situation, prediction error is maximal [3] and may function as a signal to indicate that the listener's existing internal model is no longer applicable [3,38], triggering the encoding of a new word that may later enter into the lexicon [see below for further discussion and refs 3,39] or perceptual learning if the speaker systematically produces certain speech sounds with an unusual pronunciation [40]. By this view, opposite neural effects of prediction strength during perception of matching and mismatching second syllables arise because distinct neural processes are engaged during perception of words and pseudowords.

2. I was struck by the fact that the “sharpened signals” model fails to recapitulate unexpectedness effects or “expectation suppression” (magnitude of $MM < M$ in both prediction strength “conditions”). The authors stress that the pattern simulations are more important, and that magnitude simulations “ignores contributions from neurons tuned away from the stimulus”, which they call a “questionable simplification” (l. 285-291). Yet, I was puzzled by this pattern. It seems to make the experiment unnecessary as it allows us to rule out the pure “sharpened signal” model a priori (since predictability effects are well-known to exist, even without noise, as demonstrated e.g. by the same senior author in Gagnepain et al. 2012). Does this limitation apply to earlier work from partly the same authors? Do those simulations equally predict that the sharpened signals model predicts no expectation suppression (but expectation enhancement) when stimuli are presented without noise? If so, doesn't that undermine the entire endeavor of using this “sharpened signal” model for simulations?

We thank the reviewer for drawing our attention to this point of possible confusion. As has been discussed previously (see Blank & Davis 2016 PLOS Biology; de Lange et al. 2018 Trends in Cognitive Sciences), reduced amplitude neural responses to predicted stimuli ('expectation suppression') can be explained either by sharpened signal or prediction error accounts. Thus, univariate analyses alone cannot easily distinguish between accounts and therefore in our study we consider the multivariate analyses to be most informative.

Nonetheless, we present the univariate results to offer a more complete account of our data. In the previous version of the manuscript, our univariate simulations produced different outcomes for sharpened signal and prediction error computations because we only analysed activity from model units tuned to the presented stimulus (heard speech segments). The reason for this simplification is because inherent in the sharpened signal computation is the normalization of activity such that model activations always sum to one (since activation reflects probability distributions). Therefore, summing over units to produce a univariate measure would result in identical responses for all items and conditions.

However, we agree with the reviewer that this approach for the univariate analysis is not optimal and is potentially a source of confusion given that expectation suppression is an established finding in the literature (as the reviewer notes). In the revised manuscript, we therefore explored other means of predicting the magnitude of neural activity from sharpened signal simulations. Instead of analysing activity from model units tuned only to heard speech segments, we have considered two different linking functions that would operate on sharpened signal simulations to predict univariate neural responses: (1) we summarised model responses as the entropy over the 48 segment representations, or (2) computed the normalised sum of log-transformed probabilities. The first of these linking functions is motivated by the observation that increased uncertainty in neural responses can be accompanied by an increase in the mean and therefore overall strength of neural responses (Aitchison and Lengyel 2017 Current Opinion in Neurobiology). The second linking function is motivated by the proposal that neural responses encode log rather than linear probabilities (Pouget, Beck and Latham 2013 Nature Neuroscience). Importantly, we find that either of these linking functions allows the sharpened signal model to simulate expectation suppression. These are closely related measures since “entropy” is defined as a weighted sum of log probabilities. In the revised manuscript, we depict results when using the second linking function (sum of log probabilities; shown in Figure 1D).

For the prediction error simulations, we use a simpler linking function (summed absolute prediction error, as in previous work; e.g. Gagnepain, Henson and Davis 2012 Current Biology).

Using these linking functions for the sharpened signal and prediction error univariate simulations, we now observe reduced model responses to speech segments in strongly versus weakly predicted syllables for the Match (familiar word) condition, consistent with expectation suppression. Note however, we emphasise that the multivariate analysis is most informative for distinguishing accounts, as these methods provide a way to jointly analyse responses to expected and unexpected stimulus features (see Blank & Davis 2016 PLOS Biology; de Lange et al. 2018 Trends in Cognitive Sciences). Moreover, these multivariate analyses do not require different linking functions for sharpened signal and prediction error simulations since pattern distances can be computed directly from both computational models and neural data.

These changes have been incorporated in the Results section as well as Figure 1D and Figure S3A.

3. While I really like the idea of the result in Figure 6 (decoding signal from noise or “restored syllables”, and finding an interaction, which could only be explained by prediction errors), I find the result hard to interpret. No statistics are reported other than $P < 0.05$ (which suggests $0.01 < p < 0.05$, but is quite ambiguous). No error bars are visualized. And as far as I can see there is also no statistical evidence for a main effect of “above chance” decoding in the first place. This makes it very difficult to me to assess the strength of this evidence.

Following this comment and the similar suggestion made by Reviewer 3, we have removed this analysis and associated discussion from the revised manuscript.

4. Figure 3: Isn't it a problem that the main correlations in fMRI signal magnitude and $P(\text{Syl2} | \text{Syl1})$ are positive? The authors write that this replicates the pre-onset effects in Figure 2, but when describing these results they urge caution because these may well reflect acoustic differences in Syl1 (line 343-347).

To clarify, not all fMRI correlations in Figure 3B are positive. They are positive in bilateral middle-STG, which as the reviewer notes, may reflect acoustic differences (as we discuss in the Results section). However, in right anterior STG, the correlation is numerically negative for Match items and positive for Mismatch items. Although the negative correlation for Match items is weak, the correlation for Match items is significantly smaller than the correlation for Mismatch items in this region (surviving correction for multiple comparisons across voxels). These fMRI results converge with the MEG results, reflecting two distinct effects: The first relating to responses prior to Syl2 onset (reflecting a main effect of prediction strength irrespective of match/mismatch) and the second relating to responses after Syl2 onset (reflecting the interaction between $p(\text{Syl2} | \text{Syl1})$ and match/mismatch). These findings are entirely in line with prediction error computations.

5. The authors do not report any error bars on any of the time courses. It would be helpful and important to visualize not just means but also consistencies/variability.

We now include error bars for graphs where two time-courses are displayed (e.g. Figure 4B in manuscript). For graphs that depict four conditions (e.g. Figure 4C in manuscript), we find that adding error bars leads to a cluttered appearance that negatively impacts clarity. Therefore, in these cases we do not include error bars. Note however that inset bar graphs in these figures depicting time-averaged signal in temporal clusters do provide error bars throughout. We're also mindful that – for repeated measures data such as that reported here – there's no single error bar which supports statistical comparisons between conditions (which depends on between condition variance), and comparisons with zero (which depends on between subject variance) – i.e. the difference sources of variance considered in paired t-tests, and one-sample t-tests. Moreover, neither of these error bars account for the correction for multiple comparisons (across timepoints) that have been applied here for the time-course plots. For these reasons, we have chosen to indicate the statistical significance of main effects and interactions in these time-course plots; we indicate the significant timepoints that exceed a Family Wise Error corrected level but also timepoints that only reach an uncorrected level. This provides complimentary information about effect consistency and variability.

6. A schematic of the modelling steps (extraction of segments, acoustic similarities to get likelihoods, use of lexical frequencies to get prior) would be very helpful because it is currently quite tricky to follow all details, and these aren't standard probabilities (as standard word surprisal from a language model might be).

We have now added a new supplementary figure (Figure S6) in the manuscript to provide a more detailed depiction of our computational simulations (see also our responses to a

related comment by Reviewer 1). We have also added text in the Results section to explain the relationship between the reported probabilities and surprisal (page 8, line 219):

Note that $p(\text{Syl}2|\text{Syl}1)$ can also be expressed as syllable surprisal [see 22], which is equivalent to the negative log of $p(\text{Syl}2|\text{Syl}1)$.

7. Line 315-320 – why select 20 channels? Surely the results are not fully dependent on this arbitrary number? Can the authors demonstrate some robustness?

The selection of 20 channels (with the highest condition-averaged response strength, in each hemisphere separately) is based on our experience of MEG analysis from previous work (Sohoglu and Davis 2020 *eLife*). This is a data-driven method to select channels responsive to speech. Importantly, because we average over conditions for the purpose of channel selection, this method of channel selection is also statistically unbiased (Kriegeskorte et al. 2009 *Nat Neuro*). Below in Figure R2 we demonstrate that essentially the same results are also obtained when using 10 and 40 channels.

Note that the selection of 20 channels only concerns the univariate analysis. Multivariate MEG analyses are conducted over all 204 gradiometer sensors. Nonetheless, in the revised manuscript we now mention that our univariate results are robust to how many channels are selected (Methods section page 39, line 1097).

Figure R2- Alternative version of Figure 2A from the manuscript. Traces indicate the RMS of the MEG signal across “speech-responsive” sensors in the left hemisphere, time-locked to the onset of the second syllable (Syl2). Panel A shows data when selecting 10 sensors with the highest response strength. Panel B shows data when selecting 40 sensors with the highest response strength. Thick horizontal bars at the bottom of the graph indicate timepoints when statistical contrasts were significant at a clusterwise threshold of $p < .05$ FWE corrected (thin horizontal bars show significance at an uncorrected $p < .05$ threshold). Red bars are for the main effect of congruency, magenta for main effect of prediction strength and cyan for the interaction.

8. Phonetic RDMs are based on Levenshtein distances between phonetic descriptions. Figure 4B suggests that the RDM still contains “enough” information, but it seems like quite the simplification. Why not use acoustic distances as the authors use in their simulations? Or is that very different? (Just a clarification question).

Based on the reviewer’s suggestion, we have conducted a multivariate analysis of the MEG data based on acoustic distances. Although this analysis can successfully measure neural representations of the first syllable, it is less successful for the critical second syllable (see Figure R3 below). One factor that may contribute to this is the challenge of temporally aligning fine-grained acoustic information between items (a challenge that is circumvented with the categorical phonetic representations that we currently use). This precludes us from examining condition-wise differences using acoustic distances. We have followed with interest previous discussions between Di Liberto et al (2015, Current Biology), and Daube et al (2018, Current Biology) concerning the sufficiency of acoustic vs phonetic representations of speech in neural responses but do not intend the present manuscript to contribute to this debate.

Figure R3- Alternative version of Figure 4B from the manuscript. Spearman correlation between acoustic dissimilarities and MEG pattern distances at each timepoint relative to the onset of the second syllable. Acoustic dissimilarities represent the Euclidean distances between spectra of different items during the first syllable (Syl1; in beige) and the second syllable (Syl2; in red). Horizontal bars indicate timepoints when correlations were significantly different from zero (thick lines show corrected significance at $p < .05$ clusterwise; thin lines show uncorrected significance at $p < .05$).

Reviewer #3:

Let me start by saying that although I am reasonably fluent in EEG, which carries over to MEG somewhat, I have not carried out an MEG or fMRI study, so I cannot comment on the soundness of the use of these tools.

This study builds on the authors' work exploring the neural coding of speech, with a focus on whether anticipatory processing is most accurately characterized as sharpening an

expected signal or as reducing prediction error. The two alternatives are difficult to distinguish because they often lead to the same perceptual outcome (data pattern). Across two experiments, the data provide the most compelling case for prediction error to date. Further, they provide insight into its time course and the brain regions involved. This is an important study. Suggestions below are intended to increase the impact of the work and improve clarity.

1. As written, the manuscript is very much an empirical study that adjudicates two hypothesized mechanisms. What are the theoretical implications of the results? The authors touch on these a bit, but I think far more could be done to link the findings to theory. It has been a challenge to link neural data with other mechanistic models of word processing in a substantive way. The current data can begin to place truly neurally plausible constraints on architecture. How might local suppression be implemented and how does it interface with other mechanisms involved in recognition (e.g., syllable 1 activation). A time-course account over the two-syllable stimuli would be welcome. Discussion of how this account extends beyond words to sentences and discourse would help expand the explanatory significance of the prediction error account. Ultimately we want an account that spans many levels of analysis. These data have the exciting potential to start bridging levels.

We thank the reviewer for their positive assessment of our work. In the present study we focus on processing of the second syllable since this is the key moment in the speech signal when our experimental manipulations take effect and that dissociate sharpened signal and prediction error accounts. While there would also be opportunity to explore the model dynamics and neural responses for the first syllable, our stimuli were not optimally designed for this because of acoustic confounds during the first syllable (as discussed in the Results section). Note that these acoustic confounds are not an issue for the second syllable because our stimuli in all four conditions contain acoustically identical second syllables. We view this as a major strength of our study in comparison with other less controlled methods (e.g. story listening paradigms) in which there are many possible acoustic and linguistic differences between more or less strongly predicted syllables.

We agree with the reviewer that future efforts should focus on extending these models e.g. from words to sentences/discourse. In the revised manuscript, we now discuss this in the Discussion section (page 32, line 880) although we refrain from providing extensive discussion since this issue goes beyond the scope of the current study:

The current study follows previous work that also combined computational models and neuroimaging to test sharpened signal and prediction error accounts [18,21]. As has been commented on elsewhere [64], computational models provide the most reliable way to establish the predictions of a particular theory. This observation also applies to the goal of distinguishing sharpened signal and prediction error theories. For example, opposite outcomes are observed in simulated prediction error representations depending on the level of sensory degradation [see refs 18,21 and also computational simulations in Figure S3 and Methods section]. While this result is clearly apparent in simulations, it was not previously obvious from verbal descriptions of prediction error theories. In the current study, we also see opposite

outcomes in simulated prediction errors depending on whether predictions match or mismatch sensory input during word and pseudoword processing, respectively. Through computational simulations, we could establish that this is a unique signature of prediction error representations during perception of words and pseudowords and furthermore show that this signature is observed in neural responses.

Importantly, our simulations also provide an explanation for why prediction strength and match/mismatch combine interactively to influence prediction error representations (see discussion above on why prediction error representations are enhanced by prediction strength when predictions mismatch speech input). However, future work is needed to develop a more complete model of prediction error processing [for examples of other ongoing efforts, see refs 65–67]. For example, in the present model implementation, predictions for upcoming speech sounds are subtracted from the input to compute prediction errors but these segmental predictions are never used to update higher-level lexical predictions. Rather, lexical-level predictions are computed afresh as each new segment is presented (by combining word priors and likelihoods using Bayesian inference). A more complete model would update predictions via prediction errors. It would also operate on real (i.e. acoustic) speech input or connected speech without explicitly requiring word onset locations. We note with interest the success of recent computational models which operate on acoustic speech inputs and are used to predict neural data [68,69]. We hope that the present results can constrain future iterations of these models.

2. Part of achieving the above could include making the model more prominent. It is the source of the predictions and the cornerstone of the study, yet its presentation is relegated to the Method section in a handful of paragraphs. If we are to take it seriously, integrate it into the discussion and expand its description. If the authors are not committed to the implementation (e.g., they view it as toy model or no more than a proof of concept), then this position should be stated. I view this as one of the strengths of the paper, which appears on their past work.

We thank the reviewer for this comment. We now make the modelling more prominent in the revised manuscript by implementing the following changes. Firstly, we have expanded the caption for Figure 1 to explain our computational simulations more fully. We have also added additional information on the model in a new supplementary figure (Figure S6) to illustrate the other operations performed by the model.

Secondly, we provide additional discussion of why prediction strength and congruency interactively influence simulated prediction error representations (see our response to Reviewer 2, comment 1). We believe this change helps not only to better convey the model but also to provide a mechanistic explanation of the central result of our study.

Finally, we provide further discussion of the modelling approach, including strengths, limitations and future extensions (see our response above to the reviewer's first comment).

3. Place the noise data in a separate manuscript, maybe one in which snr is manipulated and modeled. The data and their lengthy discussion are intriguing, but in contrast to the

no-noise data, their interpretation is much more speculative and somewhat off topic. They are not needed to support the authors' case.

In the revised manuscript, we follow the reviewer's suggestion and have removed this analysis and associated discussion.

4. The terse writing in the Method section is frustrating when trying to understand what was done or how it was done. As an example, only by referring to Figure 1 was I able to understand stimulus construction across conditions. The modeling paragraphs are challenging as well.

In the revised manuscript, we have expanded the caption for Figure 1 in the manuscript to explain our computational simulations more fully. We also provide additional information on the model in a new supplementary figure (Figure S6) to illustrate the other operations performed by the model. We hope this change has improved the clarity of our stimuli and modelling methods.

5. Add captions to the figures in the supplement.

We had submitted the supplementary figure captions as an attachment using the journal's editorial manager, but we think they were not included in the PDF shared with reviewers. We apologise and in our revised submission we rectify this issue.

6. Figure 1 is extremely helpful (the figures are very clear overall), but the illustrative example of "bingo" in the text and figure is confusing. The first syllable should be "BIHNG" not "BIHN" (ARPABET spelling), right?

In the previous version of the manuscript, we used an orthographic transcription system for depicting the individual syllables in Figure 1A (not ARPABET) and a custom phonetic transcription system for Figure 1B and C. To improve clarity, in the revised manuscript we have updated this Figure so that the SAMPA phonetic transcription is used throughout for depicting syllables and segments.

Below we include excerpts from the decision letter (in **bold**), along with our response (in plain text).

Reviewer #1:

Thank you for addressing my questions. I appreciate your efforts in clarifying the points I raised in the previous manuscript.

We thank the reviewer for their positive and constructive comments, which we believe have helped to strengthen the manuscript.

Reviewer #2:

It was a bit hard to review all the revisions because the changes were not tracked in the PDF. I would like to thank the authors for the effort, my comments and questions have been sufficiently addressed.

We thank the reviewer for their positive and constructive comments, which we believe have helped to strengthen the manuscript. We are sorry that the tracked changes did not remain in the PDF that was circulated.

I do have some conceptual reservations regarding the flexibility of the simulations with the choice of a linking function, and what this means for the strength of the argument of the simulations in this paper and in prior papers from the same group, but I don't see it as a deal breaker for publication.

We agree that there are challenges when selecting the appropriate linking function for the univariate simulations. However, in our manuscript we stress that the multivariate results are most informative for distinguishing accounts. This multivariate analysis provides a way to link the model with the data in a simpler and more transparent fashion than the univariate analysis. We see this as the major strength of our study that we argue addresses the reviewer's concerns.

I also agree with Reviewer 1 about the distinction between prediction error and sharpened signals. The authors rebut that some models (like classic interactive activation models) only predict sharpening to occur. However, if I understand correctly, the opposite does not seem to be true: most cortical predictive coding theories would predict both sharpening and 'prediction error' to occur. This potentially undermines the distinction the authors are after. I leave it up to R1 what to make of their response and revisions.

In our manuscript, we have been careful in framing the goal of the study. That is, we sought to distinguish between sharpened signals and prediction errors (rather than between interactive-activation and predictive coding accounts). Our results provide decisive evidence in support of prediction errors. In our Discussion, we acknowledge that predictive coding accounts incorporate both sharpened signals and prediction errors, and we suggest future avenues of investigation to reveal both forms of representation.

Reviewer #3:

The authors were responsive to my feedback. The revision is now a more tightly focused study. The amount of data presented is still large but now manageable. Although the writing is dense, it is clear and the authors do a good job of making the study accessible to a larger readership. What makes this paper such a nice piece of work is the combination of the modeling and the experiments. By elevating the model in the revision in multiple places, their synergy is now evident, and the study takes on added theoretical significance. The model makes explicit pseudoword predictions, and only by analyzing its performance can one understand the cause of the unintuitive predictions of the predictive error account. The empirical data support these predictions across multiple, complementary analyses. The field of cognitive modeling is replete with examples just like this, and they consistently reinforce the value of modeling in providing insight that leads to substantive advances. Acknowledgement of the limitations of the model and the use of pseudowords (they surely engage or mechanisms) is welcome.

We thank the reviewer for their positive and constructive comments that have led us to expand on the modelling in the manuscript. We believe this has helped to strengthen the manuscript.

One oddity to consider addressing before publication (not a requirement): Frequentist statistics are used to support inferences in various places. In light of the sophisticated acoustic modeling, stimulus construction and data processing techniques throughout the manuscript, this is incongruous. The authors are candid about weaker effects; p-values are a less-than-ideal means of conveying their strength.

We believe that our study provides robust statistical evidence for the claims made. This evidence is provided by two separate and convergent experiments. The weaker effects we believe the reviewer refer to concern one specific result within the fMRI multivariate analyses. But this result is the outcome of a secondary (follow-up) test following a statistically robust interaction.